# Imbalanced segregation of recombinant haplotypes in hybrid populations reveals inter- and intrachromosomal Dobzhansky-Muller incompatibilities

Juan Li [1,2,3]*, Molly Schumer [4], Claudia Bank [1,2,3]*

1 Institute of Ecology and Evolution, University of Bern, Bern, Switzerland, 2 Gulbenkian Science Institute, Oeiras, Portugal, 3 Swiss Institute for Bioinformatics, Lausanne, Switzerland, 4 Department of Biology, Stanford University, Stanford, California, United States of America

* juan.li@iee.unibe.ch (JL); claudia.bank@iee.unibe.ch (CB)

**Data Availability Statement:** All simulation and analysis code is archived at doi: 10.5281/zenodo.6334596.

## Abstract

Dobzhansky-Muller incompatibilities (DMIs) are a major component of reproductive isolation between species. DMIs imply negative epistasis and are exposed when two diverged populations hybridize. Mapping the locations of DMIs has largely relied on classical genetic mapping. Approaches to date are hampered by low power and the challenge of identifying DMI loci on the same chromosome, because strong initial linkage of parental haplotypes weakens statistical tests. Here, we propose new statistics to infer negative epistasis from haplotype frequencies in hybrid populations. When two divergent populations hybridize, the variance in heterozygosity at two loci decreases faster with time at DMI loci than at random pairs of loci. When two populations hybridize at near-even admixture proportions, the deviation of the observed variance from its expectation becomes negative for the DMI pair. This negative deviation enables us to detect intermediate to strong negative epistasis both within and between chromosomes. In practice, the detection window in hybrid populations depends on the demographic scenario, the recombination rate, and the strength of epistasis. When the initial proportion of the two parental populations is uneven, only strong DMIs can be detected with our method unless migration prevents parental haplotypes from being lost. We use the new statistics to infer candidate DMIs from three hybrid populations of swordtail fish. We identify numerous new DMI candidates, some of which are inferred to interact with several loci within and between chromosomes. Moreover, we discuss our results in the context of an expected enrichment in intrachromosomal over interchromosomal DMIs.

## Author summary

Genetic incompatibility in the form of (Bateson-)Dobzhansky-Muller incompatibilities (DMIs) is an important component of reproductive isolation between species. However, the evolutionary role of DMIs during the process of speciation is contentious. DMIs occur

**Funding:** CB was supported by ERC Starting Grant 804569 - FIT2GO, MS & CB was supported by HFSP Young Investor Grant RGY0081/2020, CB received support from EMBO Installation Grant IG4152. MS was also supported by Pew, Searle, and Sloan fellowships (Pew Charitable Trusts, (https://www.pewtrusts.org/en/), Searle Scholars Program (https://www.searlescholars.net/) and Alfred P. Sloan Foundation (https://sloan.org/)). The funders had no role in study design, data collection and analysis, decision to publish, or preparation of the manuscript.

**Competing interests:** The authors have declared that no competing interests exist.

when two or more genetic variants interact to reduce their carrier's fitness. Once recombination combines incompatible variants in hybrids, selection acts to remove these variants from the population. One step towards addressing the evolutionary role of DMIs is to quantify the prevalence of DMIs in incipient and hybridizing species. Here, we present statistics that are sensitive to the resulting recombinant imbalance and that can indicate the location of DMIs in hybrid genomes in various demographic scenarios. We use simulations to show that the time window during which a DMI is detectable depends on its genomic location, the severity of the DMI, and the population's demography. Importantly, our statistic distinguishes genetic associations arising due to physical linkage from those arising due to gene interactions, which allows for the inference of both inter- and intrachromosomal DMIs. Applying our statistics to three hybrid populations of swordtail fish, we confirm previously known DMIs and identify new candidate incompatibilities.

## Introduction

Hybrids between diverged populations often suffer from a fitness disadvantage caused by genetic incompatibilities between two or more loci, called (Bateson-)Dobzhansky-Muller incompatibilities (DMIs). As a result, hybrids may be less fit than their parental species, inviable, or sterile (reviewed in [1,2]). DMIs are common within and between species [3,4], often uncovered in crosses between diverged populations or closely related species (reviewed in [5,6]).

Despite their biological importance, methods for detecting DMIs in natural hybrid populations tend to have low power and statistical problems. Classical genetic mapping relies on a large sample size of hybrid offspring (i.e., hundreds or thousands of individuals) or exceptionally strong selection, and is only designed to detect pairwise interactions. Examples of known hybrid incompatibilities reflect the biases expected from statistical power limited to detecting loci with strong selection against hybrids, e.g., in the case of malignant melanoma that reduce viability in two pairs of *Xiphophorus* species (reviewed in [1,7]). In another study, researchers mapped multiple non-independent incompatibilities contributing to male sterility in a house mouse hybrid zone [8].

Another commonly used approach to infer DMIs from hybrid genomes is to search for unexpected statistical associations (linkage disequilibrium; LD) between physically unlinked regions of the genome. Studies of DMIs that use LD signals between physically unlinked loci (interchromosomal DMIs) have taken advantage of data from simulations and experiments (e.g., *Arabidopsis thaliana* [9], yeast [10], and *Drosophila melanogaster* [4]), and genomic data from hybrid populations (e.g., swordtail fish, [11,12]). Unfortunately, LD-based methods are vulnerable to high false positive rates as a function of hybrid population demographic history, especially given ongoing gene flow from source populations [12,13]. Moreover, LD-based statistics are not well suited for detecting interacting loci within chromosomes (intrachromosomal DMIs) in the absence of highly accurate recombination maps. To detect intrachromosomal interactions with LD-based methods, researchers would need to know the exact recombination rates between loci before they could distinguish expected LD caused by physical linkage from excess LD caused by DMIs. The authors of this study are not aware of any existing method that infers both inter- and intrachromosomal DMIs from hybrid genomic data.

Although intrachromosomal DMIs have not been well-studied in the hybrid incompatibility literature, several lines of evidence suggest that they should be at least as common as

interchromosomal DMIs. First, from a mechanistic perspective, gene order is not random in the genome. Genes tend to cluster on the same chromosome when they are co-expressed, co-regulated, included in the same pathway, or part of a protein-protein complex (reviewed in [14]). When sequencing and protein precipitation technology (high-C, micro-C) were combined to profile the physical interactions (e.g., interactions mediated by enhancers, insulators, cohesion binding sites, etc.) between DNA sequences, researchers found that most of the interactions were intrachromosomal, manifesting as chromosome territories and topological associated domains (reviewed in [15]). This genomic architecture of gene regulation suggests that DMIs within chromosomes should be more common than DMIs between chromosomes from first principles.

Second, from an evolutionary perspective, natural selection may favor reduced recombination between loci that are under divergent selection or involved in epistatic interactions (reviewed in [16]). Introgression is more common in high recombination rate regions of the genome where linkage disequilibrium breaks down rapidly over physical distance [17–20]. One potential consequence is that high recombination rates disassociate incompatible loci more quickly, whereas tight linkage may facilitate speciation in the face of gene flow (gene coupling, e.g., [21,22]; inversions, e.g., [23]). In line with this argument, theoretical studies indicated that hybrid populations isolated from their parent populations are most likely to develop into isolated species (homoploid hybrid speciation) with intermediate recombination rates between DMI loci [24]. Given the predicted importance of genome organization in adaptation and speciation, tools to identify intrachromosomal DMIs are needed to understand the architecture of reproductive isolation between species.

Here, we present a new statistical approach to close this key knowledge gap and to infer both inter- and intrachromosomal DMIs. We report that the imbalanced haplotype frequencies caused by selection against DMIs can lead to a unique pattern of LD and heterozygosity. We capture these patterns in two statistics that we name $X(2)$ and $\Delta D2$. We show that sensitivity and specificity of detecting DMIs are high in a range of demographic scenarios and apply our statistics to infer candidate DMIs from three swordtail fish hybrid populations. Based on this analysis, we find that putative DMIs are widespread and complex across hybrid populations of swordtail fish.

## Results

### Reduced variance in two-locus heterozygosity indicates DMIs

Heterozygosity is the expectation of the number of heterozygous loci per individual across $n$ loci, $K$, where $K \in [0, 1, 2 \ldots n]$. The variance in $n$-locus heterozygosity in this study is the variance of $K$, which can be used to describe the overall extent of linkage disequilibrium (reviewed in [25]). In the presence of population structure, the normalized deviation of the variance of $K$ at $n$ loci from its expected value under panmixia and neutrality, $X(n)$, is elevated ([26–28], see Materials and Methods). That is because with population structure, individuals breed locally, which increases the proportion of groups of homozygous individuals while maintaining polymorphism globally. This, in turn, increases the variance of heterozygosity as compared to the panmictic expectation. Here, we demonstrate that a pairwise DMI, where $n = 2$, changes $X(2)$ at DMI loci to negative values in a hybrid population, i.e., it creates a lower-than-expected variance in two-locus heterozygosity. Importantly, this statistic distinguishes LD that occurs due to physical linkage from the specific association of alleles that is caused by selection against a DMI.

To first understand the dynamics of $X(2)$ in the simplest scenario without having to consider dominance of direct selection and epistasis, we used a haploid toy model of a DMI with

recombination [29,30]. Consider a haploid panmictic population with two biallelic loci, **A** and **B,** with alleles $a$, $A$, and $b$, $B$, where $aB$ and $Ab$ represent the parental haplotypes and a hybrid incompatibility exists between alleles $A$ and $B$. We follow the haplotype frequency dynamics in a model with discrete non-overlapping generations, ignoring genetic drift and assuming that recombination occurs with probability $c$. We denote the fitness of the haplotypes by $w_{ab} = 1$, $w_{Ab} = 1+\alpha$, $w_{aB} = 1+\beta$, $w_{AB} = (1+\alpha)(1+\beta)(1+\gamma)$ (S1 Fig). Throughout the paper, we assume that $\alpha$ and $\beta$ are small positive numbers, representing a weak beneficial effect of $A$ and $B$ (following [24]), and that $-\gamma \gg \alpha,\beta$ (the incompatibility is much stronger than directional selection). We use this toy model to analyze the dynamics of $X(2)$ and $D2_{ik}$ (defined below) upon formation of a new hybrid population composed of a fraction $f$ of $aB$ parental individuals and a fraction of $1-f$ of $Ab$ parental individuals (i.e., $f$ denotes the initial admixture proportion).

The toy model highlights unique features of haplotype frequency dynamics in new hybrid populations in the presence of a DMI (see also [24] for the corresponding diploid model). Recombinant haplotypes $ab$ and $AB$ appear from the second generation onwards. At first, their frequency sharply increases due to recombination (S2 Fig). Then strong selection against the incompatibility sets in that depletes the $AB$ haplotype and gives the $ab$ haplotype a marginal advantage because offspring of $ab$ individuals never harbor the incompatible alleles (S2 Fig). Thus, the strong incompatibility drives haplotype frequency dynamics after recombinants appear.

The two-locus heterozygosity is defined as the sum of heterozygosity over two loci, $h_A+h_B$, where $h_A = 1 - p_a^2 - p_A^2$, $h_B = 1 - p_b^2 - p_B^2$ ([31]). The expected variance in two-locus heterozygosity is $h_A+h_B-h_A^2-h_B^2$ assuming binomial sampling (see Materials and Methods). The deviation of the variance in two-locus heterozygosity from its expectation under linkage equilibrium in this model is $\Delta_{var} = 2\Sigma_i \Sigma_k(g_{ik}^2-p_i^2p_k^2)$, where $p_j$ denotes the frequency of allele $j$ and $g_{ik}$ denotes the frequency of haplotype $ik$, and where we define $D2_{ik}:=g_{ik}^2-p_i^2p_k^2$ for $i\in\{a, A\}$, $k\in\{b, B\}$ (see Materials and Methods). The normalized version of $\Delta_{var}$ is $X(2) = \Delta_{var}/exp(Var)$. In a newly-formed panmictic hybrid population, $X(2)$ is initially positive because of the overrepresentation of parental haplotypes as compared with recombinants. Importantly, $X(2)$ becomes negative if the haplotype frequencies are imbalanced, for example when a lethal DMI causes absence of one recombinant haplotype and overrepresentation of the other. Focusing on the two possible recombinants in the scenario described above, this imbalance is captured by $\Delta D2 = D2_{AB}-D2_{ab}$.

The deviation of the variance in two-locus heterozygosity, $\Delta_{var}$, can also be expressed as $\Delta_{var} = 4D(2g_{ab}+2g_{AB}-2D-1)$, where $D = g_{AB}g_{ab}-g_{aB}g_{Ab}$ and $D<0$ until the population reaches linkage equilibrium or loses the polymorphism at one of the two loci. To fulfill $\Delta_{var}<0$, the condition $2g_{ab}+2g_{AB}-2D-1>0$ must be met, which implies $g_{ab} + g_{AB} - D > \frac{1}{2} \& D < 0$. It follows that the frequencies of the four haplotypes must be $g_{ab}>g_{aB}$, $g_{Ab}>g_{AB}$ (S3 Fig), indicating haplotype AB as the most unfit haplotype.

## Unique dynamics of $-X(2)$ at DMI loci

We compared the dynamics of the $X(2)$ statistic at DMI loci between scenarios where the strength of epistasis was strong ($\gamma = -0.5$) and scenarios without epistasis (i.e., $\gamma = 0$). Without epistasis and with weak direct selection for $A$ and $B$ ($\alpha = 0.001$, $\beta = 0.002$), both the recombinant frequencies and their $D2_{ik}$ values are expected to be similar and independent of the admixture proportion, because recombinants are generated at equal proportions and because weak directional selection cannot drive significant imbalance in the recombinant haplotypes (S4A and S4B Fig). Conversely, with a strong DMI, segregation of the two recombinants is imbalanced, as is evident from the $D2_{ik}$ trajectories of the two recombinants (S4C and S4D

Fig). Without epistasis, $X(2)$ rapidly decreases and converges to zero over time (S4E and S4F Fig). At equal admixture proportions, $X(2)$ for a DMI rapidly decreases to negative values and then slowly increases over time, reaching 0 when one of the interacting alleles is lost from the population. At intermediate recombination rates, $X(2)$ for a strongly selected DMI ($\gamma = -0.5$) attains its minimal value after around 60 generations (S4E Fig). In populations in which admixture proportions are highly skewed, negative $X(2)$ values are never observed, even when DMIs are under strong selection (S4F Fig). One exception to this observation is when the admixture asymmetry is counteracted by immigration from the minor parental population (discussed below). Overall, a DMI causes negative $X(2)$ once the recombinant imbalance due to selection against the DMI overcomes the residual linkage of parental haplotypes.

The signal of negative $X(2)$ generated by selection against a DMI is most pronounced when the admixture proportions are even. In addition, $-X(2)$ and $-\Delta D2$ reach the largest values at low and intermediate recombination rates (Fig 1A and 1B). At the same time, the lower the recombination rate between the interacting loci, the longer it takes for recombinants to emerge, resulting in a later appearance of negative $X(2)$ (Fig 2A and 2B). In theory, this phenomenon suggests that a DMI at any possible recombination distance will be eventually

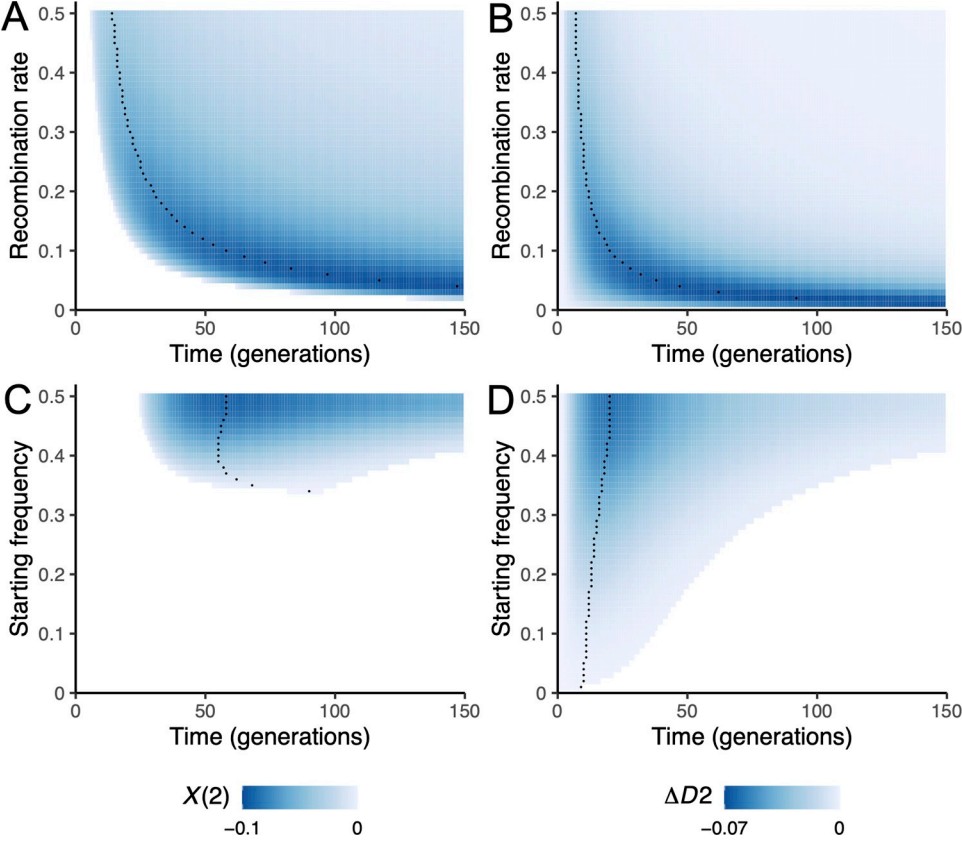

**Fig 1. Dynamics of $X(2)$ and $\Delta D2 = D2_{ab} - D2_{AB}$ in the deterministic haploid toy model for different recombination rates and admixture proportions.** The model parameters are $\alpha = 0.001$, $\beta = 0.002$, $\gamma = -0.5$. Black dots indicate the time with the largest $-X(2)$ or $-\Delta D2$ (i.e., strongest DMI signal) in each trajectory. The values of $X(2)$ and $\Delta D2$ are represented by blue color intensity. Blank areas represent either $X(2)>0$ (to the left of the blue area) or at least one locus close to fixation (minor allele frequency $<0.005$, to the right of the blue area). A&B. Both $X(2)$ and $\Delta D2$ show more extreme values at intermediate recombination rates. Here, we assumed equal admixture proportions. C&D. Negative $X(2)$ and negative $\Delta D2$ show larger values when admixture proportions are more even. In these two panels, the recombination probability between the two DMI loci is 0.1.

detectable with our approach. An exception is when two loci have vanishingly small recombination probabilities, such that direct selection and genetic drift undermine the signal of the DMI in $X(2)$, because the DMI is purged from the population (by losing one of the interacting partners) before detectable epistasis-induced recombinant imbalance appears.

Regardless of the recombination rate, our toy model suggests that detecting a DMI in an isolated hybrid population using $X(2)$ requires the initial admixture proportion to be near 0.5 (Fig 1C and 1D, minor parental frequency 0.3–0.5). Intuitively, this is because the recombinant imbalance $\Delta D2$ (which is $D \cdot (g_{ab} - g_{AB} + p_a p_b - p_A p_B) \propto D \cdot (\gamma - \alpha\beta)$) reaches its largest value at equal admixture proportions since $D = g_{AB} \cdot g_{ab} - g_{aB} \cdot g_{aB}$ is most extreme when $g_{Ab} = g_{aB} = 0.5$. As the recombinant imbalance becomes extreme, $-X(2)$ increases.

We were interested in evaluating whether $-X(2)$ is diagnostic of epistatic selection or sensitive to selection in general. Without epistasis, only strong selection in the same direction acting on both $A$ and $B$ can generate recombinant imbalance by means of Hill-Robertson interference [32,33]. To evaluate the dynamics of $-X(2)$ in this scenario, we assumed that $A$ and $B$ are under strong negative selection and combine additively, such that haplotype $AB$ is the least fit (but without epistasis). Also here, recombinant imbalance results from selection differences between $AB$ and $ab$ (S1 Fig), which cause negative $X(2)$ (S5C Fig) and $\Delta D2$. However, even with strong selection $X(2)$ remains negative only for a few generations at the very early stage of hybridization (before generation 10, S5C Fig). With a DMI, negative $X(2)$ persists for much longer. The stronger the DMI (i.e., the larger the epistasis parameter $\gamma$), the larger $-X(2)$ is observed (after generation 10, S5A Fig). Altogether, non-epistatic strong selection results in less extreme values of $-X(2)$ and $\Delta D2$ that occur–and fade–much earlier than in the presence of a DMI (S5 Fig).

## Migration can narrow or widen the detection window

Depending on the specific demographic scenario, we find that migration from one or both parental populations can increase the time during which $X(2)$ indicates a DMI (Fig 2). If the minor parental population contributes most migrants, incompatible haplotypes will be continually formed by recombination between resident and immigrant types. Therefore, the DMI persists in the hybrid population for a longer time period, which makes it possible to use $X(2)$ to detect DMIs in scenarios of uneven admixture proportions (Fig 2A and 2B). We note that immigration of the major parental population, in turn, reduces our ability to detect a DMI (Fig 2C). When immigration occurs from both parental populations, the recombinant imbalance can remain in the population for a very long time (Fig 2D). Here, depending on the parameters, a migration-DMI-drift balance can be reached which would essentially make the DMI detectable for as long as the demographic scenario persists.

## In simulated data $-X(2)$ robustly indicates intra- and interchromosomal DMIs

To demonstrate that detection of intrachromosomal DMIs in genomic data is feasible, we used simulations with SLiM ([34], Fig 3). Throughout this paper, we implemented "recessive" DMIs in our model by assuming that individuals that carry a single copy of two incompatible alleles do not suffer from reduced fitness. We simulated evolution of a chromosome in a hybrid population of 5000 individuals that is generated by a pulse of admixture between two parental populations at equal frequencies. Two interacting loci generating a DMI were located at 10 centimorgan (cM) and 20 cM such that the recombination probability between the two loci was 0.1. For these parameters, the deterministic trajectory indicated that a DMI should be detectable within a time window that spanned from generation 25 to 509, with the strongest

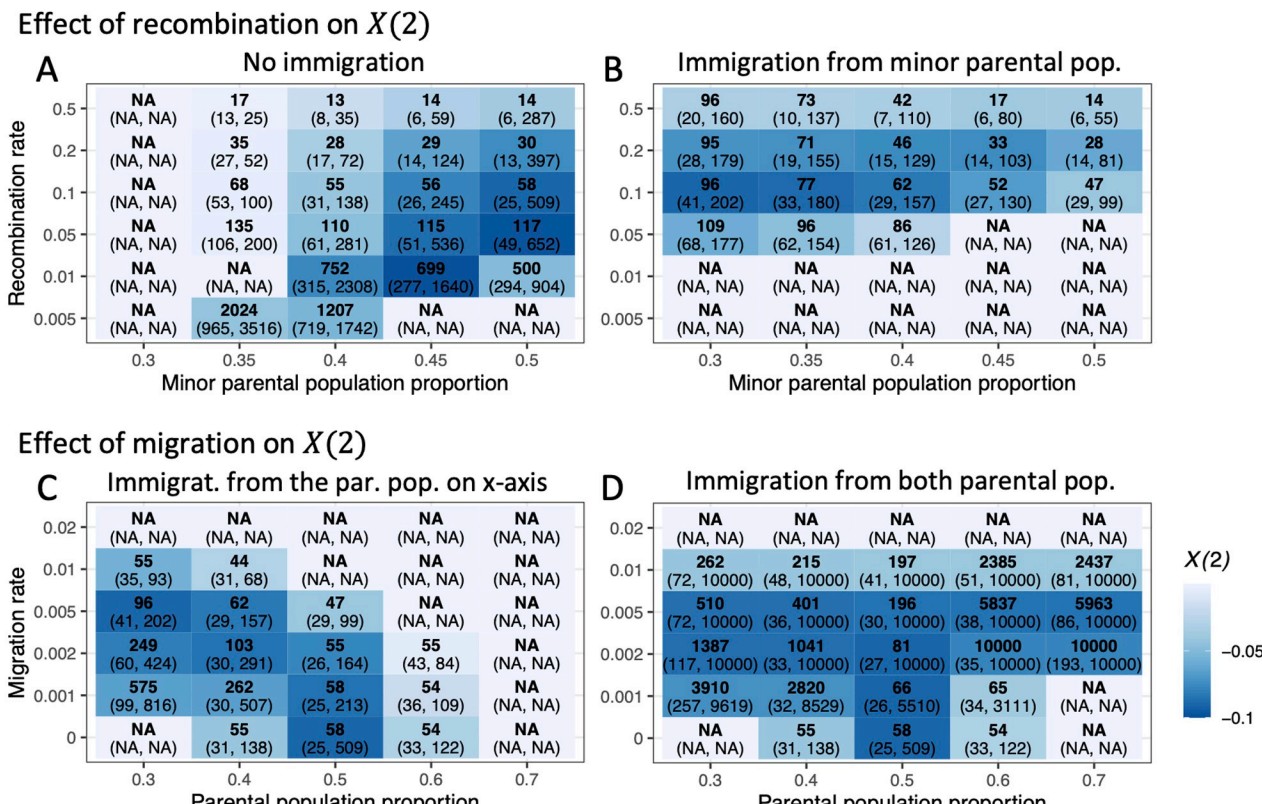

**Fig 2. The time window of negative $X(2)$ for a DMI in the deterministic toy model.** Generations with the largest $-X(2)$ values are indicated in bold. Time windows for which $X(2)$ is negative for a DMI pair are shown in parentheses. Blue color intensity indicates values of the largest $-X(2)$ observed over that interval. The model parameters used here are $\alpha = 0.001$, $\beta = 0.002$, $\gamma = -0.5$. All trajectories were traced until one locus was near monomorphic (minor allele frequency <0.005 or up to 10000 generations). We considered two parameter sets. One is a range of admixture proportions (x axis) and recombination rates (y axis) without immigration (A) or with immigration from the minor parental population ($m = 0.005$, B). A. The DMI left the strongest signal in $X(2)$ for intermediate recombination rates, and the signal was not detected until later generations when recombination rates were lower and admixture proportions were more uneven. B. Immigration from the parental population can narrow or widen the detection window. The other parameter set is a range of admixture proportion (x axis) and immigration (y axis) from one (asymmetric, C) or both parental populations (symmetric, D) when the recombination probability is 0.1. C. Immigration from one parental population can narrow or widen the detection windows. D. Immigration from both parental populations can maintain the signal of long-term DMIs. Note that the observed asymmetry in D stems from the effect of direct selection.

signal (i.e., largest $-X(2)$) at generation 58 (Fig 2A). Fig 3 shows the distribution of normalized $D$, a commonly used measure of linkage disequilibrium, ($D'$, see Materials and Methods), $X(2)$ and $\Delta D2$ at generation 50. At generation 50, parental linkage has mostly broken down between pairs of distant loci, resulting in a peak at $D' = 0$. $D' \neq 0$ only within small genetic distances (Fig 3A and 3D). Moreover, the distribution of $X(2)$ across all pairs of loci along the simulated chromosome has decreased to near 0 by generation 50 (resulting in a peak of $X(2)$ at 0, Fig 3B). In general, negative LD ($D = g_{ab} \cdot g_{AB} - g_{Ab} \cdot g_{aB}$) correlates with positive $X(2)$ (Fig 3C). However, the selected pair of loci can be pinpointed by scanning for negative $X(2)$. Negative $D'$ is also observed between the DMI loci (Fig 3D and 3E). Importantly, whereas the $D'$ signal is consistent with increased LD–that may stem from residual linkage, population structure, or epistatic interaction (Fig 3D)–-$X(2)$ is specifically associated with unexpected recombinant imbalance ($\Delta D2$) caused by the DMI (Fig 3E and 3F). Interchromosomal DMIs show a similar pattern (S6 Fig). As predicted by the toy model (Figs 1 and 2), interchromosomal DMIs show the recombinant imbalance earlier than intrachromosomal DMIs (Figs 2 and S6).

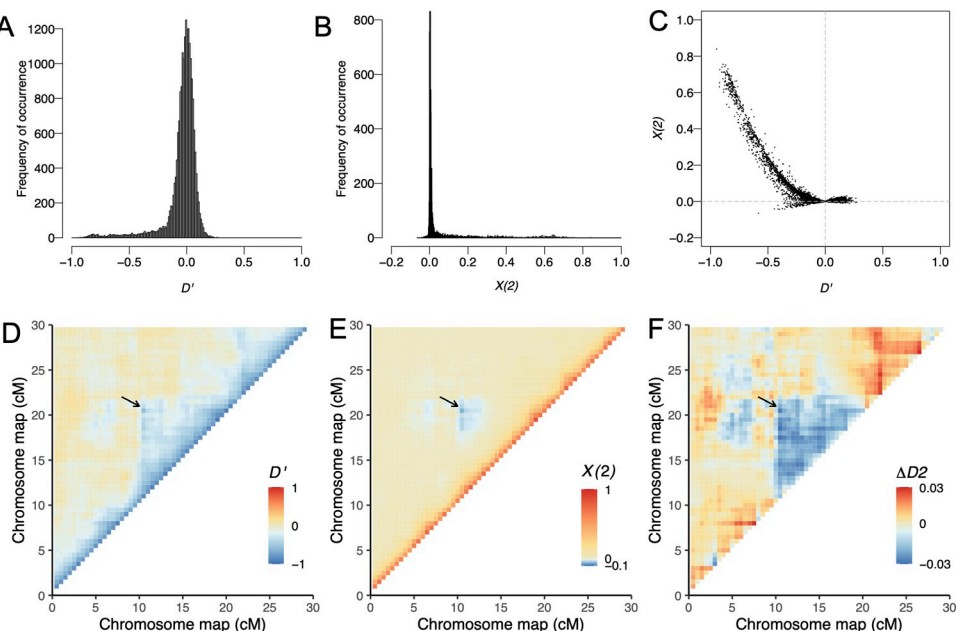

**Fig 3. Negative $X(2)$ indicates an intrachromosomal DMI in simulated data.** We show the distribution of $X(2)$, $D'$, and $\Delta D2$ of a chromosome with a DMI pair at generation 50 under a Wright-Fisher model for one simulation run using SliM. The direct selection coefficients are $\alpha = 0.001$, $\beta = 0.002$. The strength of epistasis is $\gamma = -0.5$. The interacting loci generating the DMI are located at 10 cM and 20 cM, such that the recombination probability is $c = 0.1$ between the two loci. A-C. The distribution of $X(2)$ and $D'$ and their joint distribution within the chromosome. We observe negative $X(2)$ values and negative $D'$ values at the two loci involved in the DMI and in flanking regions that are tightly linked to the DMI. D-F. Heatmaps of $D'$, $X(2)$ and the $\Delta D2$ within the simulated chromosome. Only the DMI region and its 10 cM flanking regions are shown in D-F. The DMI is highlighted by a black arrow.

Since other metrics used to detect DMIs are sensitive to demographic parameters, we next used SliM to explore whether the same was true for $X(2)$. We first considered the effect of variable population size. Negative $X(2)$ values still appear when the hybrid population size is as small as 300 (Fig 4A). At smaller population sizes, genetic variation in the hybrid population is lost more quickly than for larger populations, which leads to faster purging of the DMIs and thus shortens the detection window (Fig 4A). We further considered how population expansion affects the sensitivity of the $X(2)$ statistic to DMIs. Natural hybrid populations may suffer from low fertility and high mortality at the early stages of hybridization, and the hybrid population may occupy a new niche or geographical area. As hybrid incompatibilities are purged, mean population fitness increases and the population may expand. To explore the possible effects of this phenomenon, we performed simulations assuming that the hybrid population would expand within 35 generations by arbitrarily defining a 5% growth rate per generation. We found that population expansion does not affect the detection of a DMI (Fig 4B) since it does not alter the relative genotype frequency dynamics.

We next considered continual immigration from the parental species into the hybrid population. As shown in Fig 4B, high migration rates initially reduce our ability to detect negative $X(2)$ when admixture proportions are even. However, depending on the specific migration scenario, migration can aid the detection of a DMI. Generally, migration changes the haplotype frequencies and alters the time period during which $X(2)$ is negative (Fig 4B). To quantify this effect, we studied three migration rates, no ($m = 0$), intermediate ($m = 0.005$), and strong ($m = 0.01$) migration from the minor parental population into the hybrid population, and strong migration from both parental populations under even admixture proportions. If the

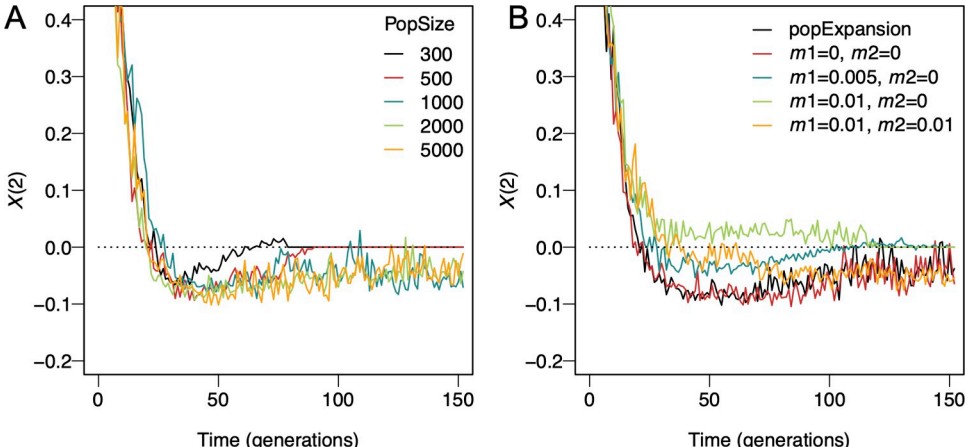

**Fig 4. Effect of demographic parameters on the dynamics of $X(2)$ for a DMI in a hybrid population.** Each line is one randomly chosen simulation run using SLiM, assuming equal admixture proportions of the parental populations and a strong DMI ($\alpha = 0.001$, $\beta = 0.002$, $\gamma = -0.5$) under a Wright-Fisher model. The recombination probability between the interacting loci was 0.1. 300 individuals were sampled in each simulation independent of the population size. Individuals that did not contain at least 10% of their genome from either parental population were excluded. Trajectories were plotted from generation 2 onwards. A. $X(2)$ dynamics with variable population size. $X(2)$ indicates DMIs when the population size is as small as 300. DMIs are purged more quickly from small populations due to genetic drift. B. $X(2)$ dynamics with population expansion and migration. Immigration rates are $m_1$ amd $m_2$ from the two parental populations. The $X(2)$ signal is only mildly affected by population expansion, low asymmetric migration, and symmetric migration from parental populations. Strong migration from one parental population disrupts DMI detection by $X(2)$ (green curve).

asymmetric migration rate is small to intermediate ($m \leq 0.005$) or migration rates from two parental populations are symmetric, our simulations suggest that migration does not impede the occurrence of negative $X(2)$ in the presence of a DMI (Figs 2C, 2D and 4B). Symmetric high migration rates could sustain long-term DMIs (Figs 2D and S7).

With strong migration ($m > 0.005$), the linkage of parental haplotypes remains strong due to the continuous reintroduction of these haplotypes (S8 Fig). In this scenario, $X(2)$ remains larger than 0 for an extended time. Thus, DMIs can only be detected with $X(2)$ at a later age of the hybrid population, if the DMI alleles have not been purged from the population due to asymmetry of admixture or immigration (Figs 4B and S8). Here, strong migration acts like strong selection for the immigrating parental two-locus haplotypes, which hides the (still existing) recombinant imbalance.

Next, to evaluate the performance of $-X(2)$ in a natural population with overlapping generations, we compared our results from Wright-Fisher simulations with those obtained under a Moran model. With overlapping generations, the emergence of negative $X(2)$ is delayed because recombinant imbalance builds up more slowly than in a model with non-overlapping generations (S9A Fig). At the same time, alleles $A$ and $B$ segregate in the population for much longer with overlapping generations than with non-overlapping generations (S9B Fig). This leads to a longer maintenance time of the interacting loci, resulting in a longer window during which $-X(2)$ can be used to detect the DMI.

Given these results, we next evaluated the sensitivity and specificity of $X(2)$ for DMI detection under a range of models using non-overlapping generations (Fig 5). We simulated four chromosomes, two of which contained the DMI loci and two of which evolved neutrally. We computed sensitivity as the proportion of simulations in which the DMI loci showed $X(2) < -0.005$ & $D' < 0$, and specificity as one minus the proportion of simulations in which a pair of

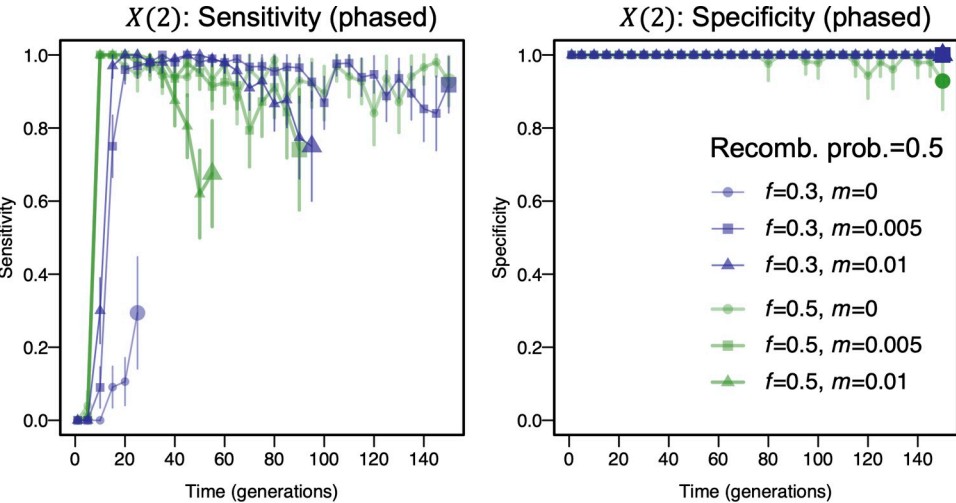

**Fig 5. Sensitivity and specificity of negative $X(2)$ for detecting a DMI, computed from 100 simulations of each parameter combination using SliM, assuming a Wright-Fisher population of 5000 individuals.** We considered two admixture proportions, $f = 0.5$ and $f = 0.3$, and three migration rates from the minor parental population: 0, 0.01, and 0.05. A strong DMI ($\alpha = 0.001$, $\beta = 0.002$, $\gamma = -0.5$) and no DMI ($\alpha = 0.001$, $\beta = 0.002$, $\gamma = 0$) were simulated under a Wright-Fisher model. The recombination probability between the DMI loci was $c = 0.5$. To exclude sampling noise, any interaction with $X(2) < -0.005$ & $D' < 0$ was taken as a putative DMI. Trajectories ended when one of the interacting loci became monomorphic (indicated by large plot markers). Three hundred individuals were sampled in each simulation. A. Sensitivity is high unless strong migration leads to swamping of the DMI alleles. B. Specificity is high under all tested scenarios when no DMI locus is present on the sampled chromosome. Vertical lines represent the 95% confidence interval (Wald test).

loci on the neutral chromosomes showed $X(2) < -0.005$ & $D' < 0$. We considered six scenarios by combining two admixture proportions ($f = 0.5$ and $f = 0.3$) and three immigration rates (0, 0.005, 0.01) of the minor parental population. We also compared the performance of $X(2)$ to the G test which has previously been used to detect DMIs [10]. The recombination probability in these simulations was set to $c = 0.5$ because the G test can only be applied to unlinked loci.

In simulations of hybrid populations with non-overlapping generations, equal admixture proportions, and no or moderate levels of ongoing migration from the minor parental population, $X(2)$ allows for detection of the DMI with high sensitivity and specificity (Fig 5, $f = 0.5$, $m = 0$ or 0.005). At high migration rates ($m = 0.01$), migration from the parental population dominates the haplotype frequency dynamics and the DMI cannot be detected with equal admixture proportions. In contrast, immigration can maintain the minor parental alleles when the admixture proportions are uneven ($f = 0.3$). With continuous input of the minor parental haplotype, the DMI remains in the population for longer, so the DMI can be detected based on negative $X(2)$ values until one of the parental haplotypes is lost. Thus, $X(2)$ can be used to detect DMIs in the case of uneven admixture proportions and intermediate migration rates, when immigration counteracts selection against the DMI in the minor parental population. In all scenarios explored here, $X(2)$ has very high specificity. In 600 simulations of a null model of neutral evolution, $X(2)$ was always greater than -0.001, where negative $X(2)$ was caused by genetic drift and random sampling of 300 individuals for the analysis. At low recombination rates, a similar pattern can be observed (S10 Fig).

We compared the performance of $X(2)$ with that of the G test, which was identified as one of the best statistics to detect interchromosomal DMIs [10]. S11 Fig shows that the G test is highly sensitive at the early stages of hybridization, but its sensitivity decreases after generation 20. Here, we also observed a tradeoff between sensitivity and specificity. Interestingly, unlike

for the G test there was no tradeoff between sensitivity and specificity for $X(2)$. We found that whereas significant hits using the G test had negative $X(2)$, the opposite was not true, especially when the parental linkage has broken down from generation 20 onwards (Figs 5, S11 and S12). Thus, $X(2)$ was sensitive to DMIs in scenarios in which the G test was not.

Our analysis above relies on the use of phased genotyping data, which can be difficult to collect in practice and can introduce inaccuracies via phasing errors. In contrast, pseudo-phased data are available for many datasets, including the data analyzed below. In this case, only homozygous loci in each individual are used for analyses where the phase can be inferred without computational phasing (e.g., genotyping data reviewed in [35]). We calculated the sensitivity and specificity for the G test and $X(2)$ using only homozygous genotypes (same ancestry state at an ancestry informative site, see Materials and Methods). The performance trends for pseudo-phased data were similar to phased data with mildly increased sensitivities of $X(2)$ and decreased sensitivities of G test (Figs 5, S11 and S13).

We next applied our detection method to simulated genomic data from a range of more general scenarios with one and two standard DMIs, and a three-locus interaction in which two DMIs had one shared interaction partner. Here, positions of the interacting loci were distributed randomly across four chromosomes (resulting in both inter- and intrachromosomal DMIs), epistatic coefficients were drawn from a uniform distribution ($\gamma \in [-1, -0.001]$), and the direct selection coefficients $\alpha$ and $\beta$ were drawn from an exponential distribution with mean 0.001. We further simulated two neutral chromosomes without DMI loci.

On neutral chromosomes, negative $X(2)$ values were always larger than -0.025 (Fig 6A). On chromosomes with DMIs, >75% of negative $X(2)$ were larger than -0.005 (Fig 6B). However, $X(2)$ values can take smaller values up to -0.12 at and close to DMI loci (Fig 6B). $X(2)$ at DMI loci tended to show extreme values (Fig 6C). Notably, under the same selection and demographic parameters, when two DMIs shared one locus (three-locus two-DMI model, see Materials and Methods), the $X(2)$ distribution at the DMI loci was slightly closer to zero than the distributions for two separate DMIs (Figs 6 and S14). In all cases, $X(2)$ distributions in both models could clearly separate DMI loci from neutral loci (Figs 6 and S14). This indicates that $X(2)$ can still detect DMIs when interaction hubs are present. When only homozygous loci (pseudo-phased data) were used for the inference (S14 Fig), the distribution of $X(2)$ values was similar but chromosomes with DMIs showed more extreme values. This is because the reduction in sampling size due to pseudo-phasing stochastically resulted in extremely low numbers or absence of incompatible haplotypes.

The proportion of pairs of loci that were incorrectly inferred as interacting partners (from here on referred to as false positive rate) varied significantly between chromosomes with DMIs and neutral chromosomes (S1–S3 Tables). For the neutral chromosomes, this proportion was almost 0 when $X(2) < -0.005$ was used as a threshold (S2 and S3 Tables) for both phased and pseudo-phased data. However, on chromosomes with DMIs, upwards of 2% of pairwise comparisons (from a total of $\binom{400}{2} = 79800$ pairs, of which 2 pairs were true DMI pairs, corresponding to 0.005% of pairs) were identified as putative DMIs based on the $X(2)$ statistic (threshold $X(2) < -0.005$) although the two loci were not involved in a DMI. Most of this signal came from loci in the direct vicinity of the DMI. As expected, both true and false positive rates on the chromosomes with DMIs decreased with a more stringent $X(2)$ threshold (S1–S3 Tables). Comparing to simulations with a single DMI, the false positive rate increased noticeably but not excessively when two DMIs were simulated (from ~1% to a maximum of 2.2% when $X(2) < -0.005$, S1 and S2 Tables), indicating some interference between DMIs. The true positive rate remained similar between one-DMI and two-DMI models (around 50%-70% when $X(2) < -0.005$, S1 and S2 Tables). Also when only homozygous loci (pseudo-phased data)

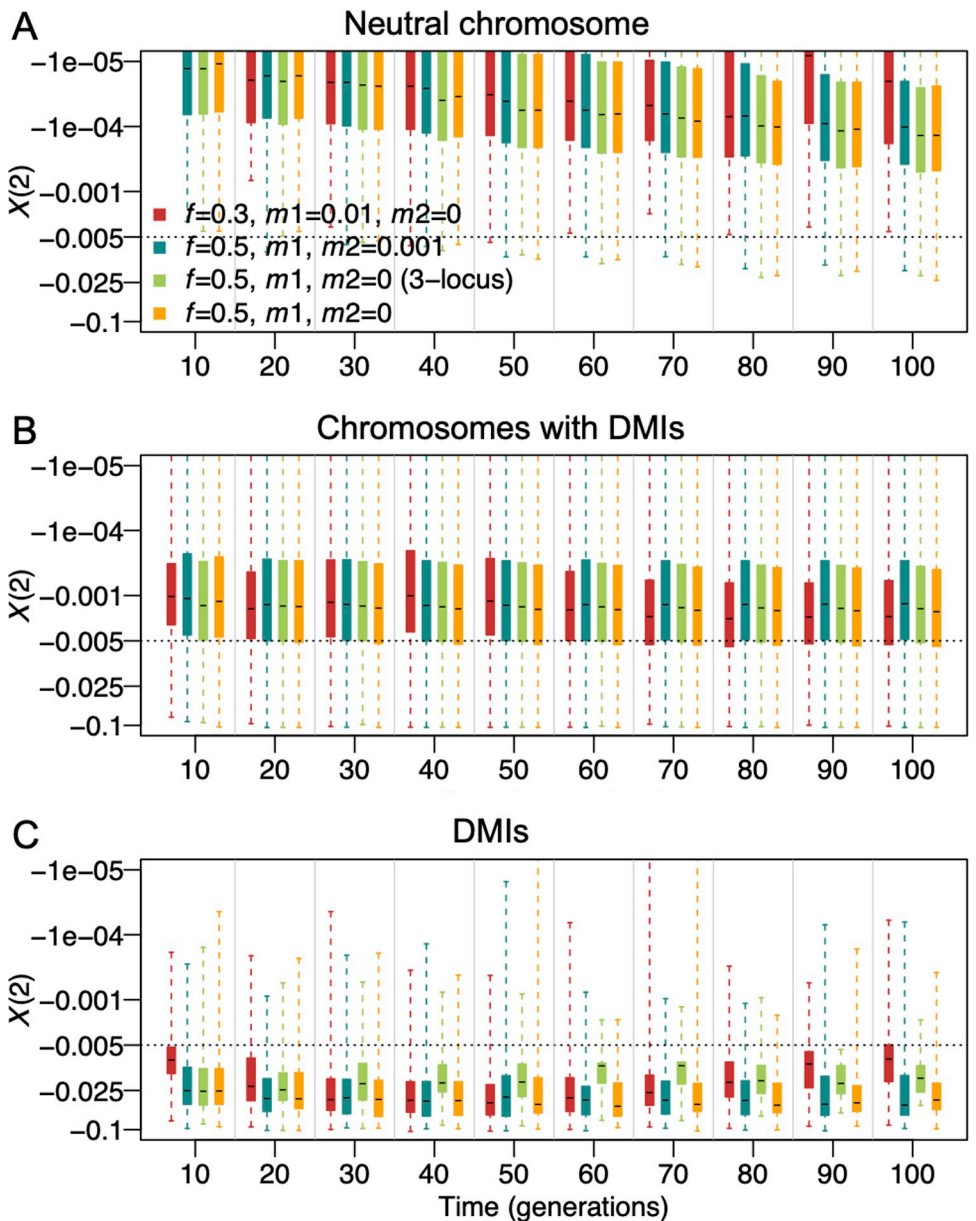

**Fig 6. Distributions of negative $X(2)$ when $D'<0$ for DMI detection under four demographic scenarios.** This figure represents two starting frequencies of the minor parental population, $f = 0.5$ and $f = 0.3$, and different immigration rates from the two parental populations ($m_1$ from minor parental population, $m_2$ from major parental population). Light green boxes show the $X(2)$ distribution for the three-locus two-DMI model (3-locus) and other boxes for the four-locus two-DMI model. DMIs are randomly distributed on four chromosomes. Direct selection coefficients of the incompatible alleles were drawn from an exponential distribution with mean = 0.001. Epistatic coefficients were drawn from a uniform distribution [-1, -0.001]. We calculated $X(2)$ statistics from a sample of two hundred individuals for each simulation. We simulated each scenario 100 times with SLiM. $X(2)$ values were summarized at three types of genomic regions, neutral chromosomes (A), chromosomes with DMIs (B), and individual DMI loci (C). Boxplots represent the interquartile range; whiskers extend to minimal and maximal values. The dashed line indicates the studied detection threshold $X(2) = -0.005$.

were used for the inference, the true positive rate remained similar, but the false positive rate was elevated to 3.6–6.1% on chromosomes with the DMIs, presumably due to stochastic absence of haplotypes from small samples (S2 and S3 Tables).

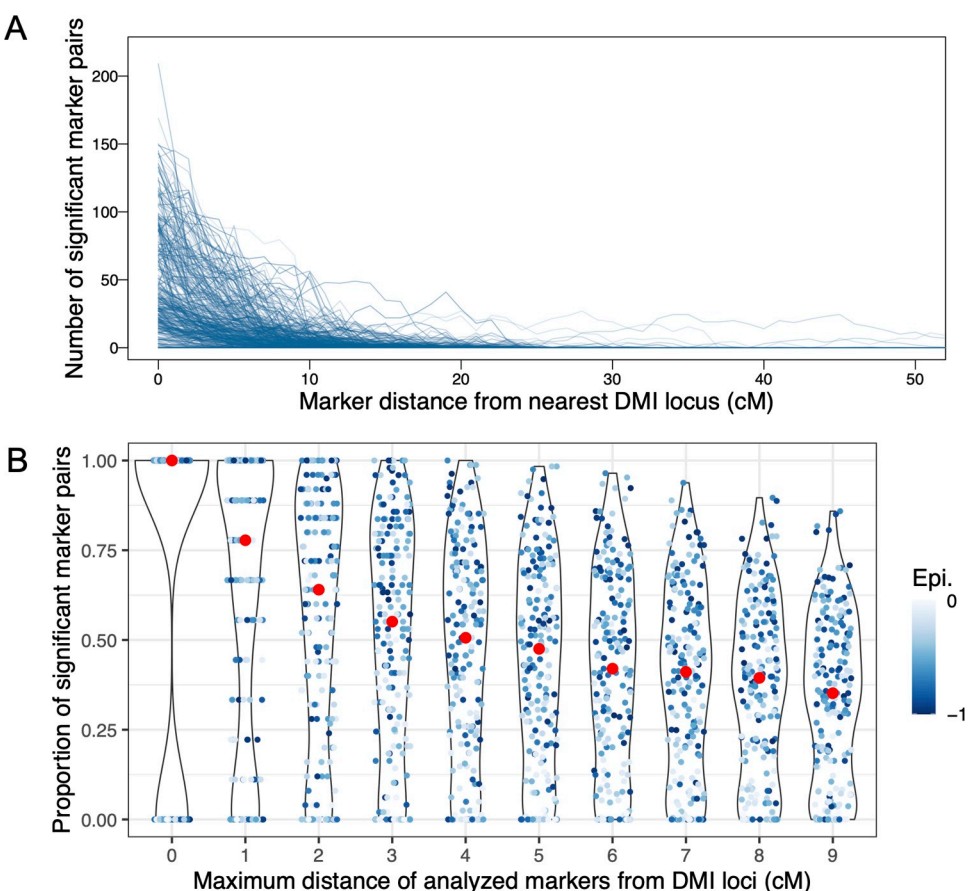

**Fig 7. Propagation of the DMI signal into flanking regions in simulated data (sampled at generation 30, significance threshold: $X(2)<-0.005$ & $D'<0$).** Data resulted from 100 simulations using SLiM, in which two DMIs (i.e., four DMI loci) were randomly distributed on four chromosomes, thus resulting in inter- and intrachromosomal DMIs. The direct selection coefficients $\alpha$ and $\beta$ were drawn from an exponential distribution with mean 0.001. The strength of epistasis was drawn from a uniform distribution, $\gamma\in[-1, -0.001]$, which is represented by the blue color intensity in both panels. To resemble the demography of the studied hybrid swordtail fish populations, the admixture proportion was set to $f = 0.3$ and the migration rate was $m = 0.01$. Two hundred individuals were sampled from each simulation to obtain the statistics. A. The number of significant marker pairs decreases rapidly with distance to the nearest true DMI locus. B. Proportion of significant marker pairs when statistics were computed for all pairs between two increasingly large regions around the true interacting DMI loci. Red dots indicate the median obtained from 100 simulations. Both weak and strong DMIs can be detected, but weak DMIs tend to leave a weaker signal that did not extend as far as the signal of strong DMIs.

The great majority of false positives as defined above were caused by loci linked to DMIs (e.g., Figs 3E and S6). Markers within 10 cM of DMI loci showed many significant $X(2)$ pairs in simulations of two DMIs that were distributed randomly across four chromosomes (Fig 7A). In the most extreme case observed in the simulations, one DMI locus was involved in significant pairwise interactions with over 200 markers of a total of 400 markers (Fig 7A). This case involved strong epistasis for both DMI pairs ($\gamma\approx-0.75$); two DMI loci that were located very close to each other on the same chromosome, where DMI 1 was between chr1:63cM and chr4:100cM and DMI 2 between chr3:98cM and chr4:92cM; the extreme signal was emitted from chr3 position 98. Thus, the extreme signal was likely caused by interference between the two DMIs, i.e., a distortion of the signal of one DMI due to selection against the other one. At generation 30, the interaction signal of the two DMIs extends along a large genomic window.

Here, the proportion of significant pairs is high within 10 cM of the true DMI loci (median ~35% vs. genomic median 1.1%, Fig 7B and S2 Table), probably caused by hitchhiking or limited sorting of large genomic blocks in young hybrid populations.

In analyses of unphased genomic data, we can only rely on information from homozygous genotypes. Therefore, we computed the same statistics for simulated data when only homozygous genotypes were considered (S15 Fig). Here, consistent with the lack of resolution suggested above, the proportion of significant pairs between DMI regions decreased more slowly with window size than when the phased data were considered (S15C Fig; see also Fig 7B). We then tested the approach with the strict filtering criteria below to the genomic data from natural hybrid populations before analysis (see below and Materials and Methods), for which a homozygous frequency of >0.05 and a frequency of heterozygotes <0.6 were required. Because most strong DMIs are (nearly) purged by generation 30, few DMI loci pass the filtering, and the signal therefore changes such that the highest proportion of significant pairs is observed for genomic windows that span several cM around the true DMI (S15B and S15C Fig). Here, only the flanking regions of strong DMIs (and not the DMI loci themselves) indicate the DMI. The proportion of significant pairs at up to 10cM distance from a true DMI locus is still higher than the genome-wide false positive rate (>35% vs. genomic median up to 7.1%, S15B Fig and S3 Table). Taken together, our analysis indicates that, although greater precision could be obtained from phased data, our method can identify selection in pseudo-phased data.

## Using $X(2)$ to detect putative DMIs in swordtail fish hybrid populations

Two sibling species of swordtail fish, *Xiphophorus birchmanni* and *X. malinche*, were reported to hybridize naturally in Mexico [11,36]. Since these hybrid populations formed recently and have been studied extensively, they represent an excellent opportunity for using $X(2)$ and $\Delta D2$ to infer DMIs from natural populations. Three hybrid populations called CHAF, TOTO and TLMC were sampled and genotyped in previous work (S4 Table). All three hybrid populations are estimated to have formed less than 100 generations ago: CHAF ~45 generations; TOTO ~35 generations; TLMC ~56 generations [7,11]. However, these admixture times may be underestimated due to ongoing migration [11, 18, 37]. The estimated proportion of the genome derived from the minor parental population was ~0.25–0.3. The minor parental species for the CHAF and TLMC populations is *X. birchmanni* and for TOTO it is *X. malinche*. All three hybrid populations are known to experience some level of ongoing migration from one parental population (CHAF, ~5% from *X. birchmanni*; TOTO, ~3% from *X. malinche*; TLMC, <<1% from either species; [7,37]; S4 Table). Under the inferred demographic parameters of these natural populations, we expect that in CHAF and TOTO, immigration from the minor parental population counterbalances the asymmetric admixture proportions such that we can infer DMIs with intermediate sensitivity and high specificity (similar to Figs 2, 5, S10 and S13).

We used $X(2)$, $\Delta D2$ and $D'$ to identify putative DMI pairs and to infer the most strongly depleted genotypes in these three populations. To roughly count the DMIs in a fashion that is comparable between populations, we sliced the genome into fragments of 1 Megabase (Mb), which we will refer to as a locus in the following analyses. To increase our confidence in the candidates, we performed bootstrapping to ensure that the locus pair was consistently inferred to have negative $X(2)$ and $D'<0$ (S5 Table, see Materials and Methods). The number of inferred DMIs is quite robust to different $X(2)$ cutoffs (from 0 to -0.015, S5 Table). A putative DMI was identified if homozygous markers at this locus showed $X(2)<−0.005$ and $D'<0$ (see Materials and Methods), based on the observation that neutral and DMI loci can be separated by negative $X(2)$ at -0.005 from simulations (Figs 6 and S14).

In total, we detected 2203, 2039 and 1442 putative DMIs in the three hybrid populations CHAF, TLMC and TOTO with an arbitrarily determined false discovery proportion <0.001 (15.2%, 12% and 6% of tests, respectively, S5 Table; see Materials and Methods). As highlighted by the simulations in the previous section, these numbers may include many genomic blocks that are close to DMIs but do not themselves carry a DMI locus (Figs 7 and S15, and S5 Table). Given the moderate false positive rate estimated from the simulations, we view this as a preliminary set of candidate DMIs for further investigation. In each population, 200–400 1Mb bins (out of the total 711 1Mb bins, S5 Table) were involved in at least one putative DMI. If the putative DMIs were randomly distributed across the genome, the occurrences of the 1Mb bins would fit a Poisson distribution ($\lambda = 2 \cdot$ number of putative DMIs / number of 1Mb bins involved in interactions). In contrast to this expectation, 1Mb bins with very small and vast numbers of putative interacting partners were greatly overrepresented among the DMI candidates (One-sample Kolmogorov-Smirnov test, $p < 2.2e{-}16$, S16 and S17 Figs). We speculate that the 1Mb bins that appear in many putative DMIs are likely to contain true DMI loci in pairwise or complex DMIs, whereas many of the 1Mb bins that occur only once or twice may be noise caused by linkage to the true DMI loci or additional background noise. For example, the noise could stem from negative $X(2)$ between two flanking regions of a DMI pair, which is weak but observable. More than 99.5% of putative DMIs are caused by a small number of 1Mb bins that are each involved in >10 putative DMIs (54 in CHAF, 61 in TLMC and 66 in TOTO). Among these putative DMI hubs, CHAF and TLMC share 8 loci, CHAF and TOTO share 2, and TLMC and TOTO share 2 (S6 Table). Altogether, these results suggest that the signal we observe is driven by a small set of loci that are involved in complex DMI interactions in the hybrid population at the sampling time point.

Although our overall false positive rate is low, given the large number of tests performed in our evaluation of the real data, we can be most confident in putative DMIs that are detected in multiple populations. Across three hybrid populations we found 17 putative DMI pairs shared between TLMC and CHAF, which are the two populations that have the same minor parental population (*X. birchmanni*) and are therefore expected to share more detectable DMIs (S7 Table). There were no shared DMI pairs between TOTO and the two other populations. The overlapping proportions are lower than expected by chance with randomly distributed DMIs per population (binominal test, $p < 0.05$). Among all putative interaction partners, 14 loci were shared by three populations, out of which four had been previously identified (S8 Table). This set of loci are exciting candidates for future work. Out of all putative interactions identified in the swordtail hybrid populations, about 4% were intrachromosomal (CHAF: 81/2203; TLMC: 67/2039; TOTO: 55/1442). Notably, none of the inferred intrachromosomal interactions overlapped between any two populations.

The small number of overlapping DMI candidate pairs between the populations most likely results from the individual demography of the hybrid populations. Firstly, as we reported above, the $X(2)$ signal of weaker DMIs and more closely linked DMIs appears later. Therefore, hybrid populations of different ages may not display the same set of detectable DMIs. Secondly, the hybrid populations have different admixture proportions and immigration rates (S4 Table). These likely result in different sets of segregating DMIs and different trajectories of purging of DMIs. For example, S18 Fig shows a putative intrachromosomal DMI in TLMC (between chr20:9M and chr20:10M). In the $\Delta D2$ heat plot of this genomic region, the candidate recombinant from TLMC (composed of *X. birchmanni* ancestry at chr20:9M and *X. malinche* ancestry at chr20:10M) also showed a weak signal of elimination in CHAF. However, in TOTO, which has opposite admixture proportions, no such recombinant imbalance is visible. Another interchromosomal DMI also has a similar pattern in TLMC and CHAF, but with less residual recombinant imbalance in the flanking regions in TOTO, where the focal locus is already fixed (S19 Fig).

In comparisons between TLMC and CHAF, which share a minor parent species, two loci stood out with respect to their many inferred interaction partners. Both chr20:23Mb (specifically, between 23Mb and 23.4Mb) and chr15:15M (specifically, between 15.6Mb and 16Mb) are putatively involved in 8 interchromosomal interactions. At both loci, alleles from *X. malinche* are incompatible with alleles at several loci from *X. birchmanni* (S7 Table). Interestingly, there was no signal of an interaction between these two loci. However, chr18:19Mb is putatively interacting with both chr20:23Mb and chr15:15Mb in both TLMC and CHAF. *X. malinche* is the major parental population in TLMC and CHAF, leading to the expectation that homozygous *X. malinche* alleles would be in the majority. In contrast to this expectation, few alleles from *X. malinche* remained in these regions when combined with several *X. birchmanni* backgrounds. In TOTO, where *X. malinche* is the minor parental population, many *X. malinche* alleles were already lost from the population or are segregating at very low frequencies. This leads to a power reduction for detecting *X. malinche*-specific interacting partners hinting at why the two strong DMI candidates may not be detectable in TOTO. For example, at locus chr20:23M, frequencies of homozygous *X. malinche* genotypes were found at a median frequency of 0.004, which is much smaller than our filtering criteria 0.05. No DMIs remained detectable in TOTO at locus chr15:15M, at which the frequencies of homozygous *X. malinche* genotypes were found at a median frequency 0.009. Together, our results point at chr20:23Mb and chr15:15Mb as strong DMI candidates at which *X. malinche* alleles are deleterious in combination with certain (and possibly several) loci from *X. birchmanni*.

In addition to comparing our candidates with a list of previously identified DMIs (S6 and S8 Tables), we specifically analyzed genomic regions involved in a well-studied DMI that is known to induce melanoma in hybrids. A DMI has been confirmed between chr5:10-11Mb (*cd97*) and chr21:16.5–17.5Mb (*Xmrk*, [7]). A previous QTL assay predicted that another gene, *cdkn2a/b* (~chr5:16-17Mb; reviewed in [38]) interacted negatively with *Xmrk*. Previous work studying the interaction between *Xmrk* and *cd97* has focused on CHAF, where melanoma frequency is high (interactions generating melanoma appear to have been purged in the other hybrid populations). Interestingly, in CHAF, it appears that both homozygous hybrid haplotypes are depleted in a part of this region. Thus, the *Xmrk-cd97* interaction could be a hybrid incompatibility in which both recombinant haplotypes are deleterious, in which case no negative $X(2)$ signal is expected (since the signal is driven by recombinant imbalance as caused by classical DMIs). Regarding the potential DMI between *Xmrk* and *cdkn2a/b*, we found negative $X(2)$ in at least two hybrid populations that support this interaction (S20 Fig and S9 Table). In CHAF and TOTO, markers linked to *Xmrk* passed the filtering criteria, but none in TLMC (S9 Table). In CHAF, $X(2)$ of 49.5% of marker pairs was negative but only 0.2% of these passed the threshold $X(2)<-0.005$. In TOTO, which is of intermediate age, 25.9% of all marker pairs showed $X(2)<-0.005$, which is much higher than the expectation obtained for a DMI from bootstrapping (median = 5.7%, S5 Table). This is consistent with our expectation of the signal of an intermediate-strength DMI that does not display strong negative $X(2)$ until later generations.

## Discussion

DMIs play an important role in maintaining reproductive isolation between species. Despite their crucial importance, existing methods to identify DMIs are underpowered and susceptible to high false positive rates. Here, we show that the deviation of the variance in two-locus heterozygosity from its expectation, $X(2)$, can allow us to identify intra- and interchromosomal DMIs in hybrid genomes. Specifically, a negative $X(2)$ statistic is indicative of the presence of a DMI, because epistatic selection transiently leads to an imbalance of the recombinant

haplotypes. A second measure that we develop, $\Delta D2$, quantifies this recombinant imbalance more explicitly. After showing the expected sensitivity and specificity of these statistics in simulations, we used $X(2)$, $\Delta D2$ and $D'$ to infer candidate DMIs from three hybrid populations of swordtail fish. Although the detection power with these statistics is strongly dependent on the age and demographic history of the hybrid population, false positive rates across simulations of various scenarios were low and are expected to be especially low among interactions detected in multiple populations. Encouraged by this result, we found that natural hybrid populations of swordtail fish harbored many potential DMIs, and our analysis detected known DMI pairs. Interestingly, putative DMI pairs tended to include the same loci more often than expected by chance, which is suggestive of the presence of interaction hubs in these data.

## DMIs cause distinctive patterns of LD and variance in two-locus heterozygosity

Linkage disequilibrium (LD) between physically unlinked loci has been frequently used to identify candidate DMIs. However, LD is influenced by many factors including selection, population structure, population differentiation, genetic drift, and epistasis. The variance in two-locus heterozygosity was first proposed as a measure of multi-locus LD [26]. Generally, LD tends to raise the variance in heterozygosity above its expected value at linkage equilibrium. This deviation from the expectation (called $X(n)$, where $n$ is the number of loci considered) was used to infer population structure within species, e.g., in natural populations of *Hordeum spontaneum* and bacteria [27,28]. These studies assumed that the focal loci are independent, and that any disruption of independence would cause LD and a positive deviation from the expected variance in $n$-locus heterozygosity (positive $X(n)$). Our study shows that selection against DMIs generates a unique pattern of LD and $X(2)$, where LD is strong but $X(2)$ becomes negative. The conditions for negative $X(2)$ are met when the imbalance happens between a pair of repulsive haplotypes (here, ab and AB) and this pair of haplotypes is in deficiency (linkage disequilibrium, $D = g_{ab} - p_a p_b < 0$).

## DMI inference with $X(2)$ does not rely on recombination rate estimates

One advantage of our statistics is that they do not rely on knowledge of recombination rates, which are often unknown or estimated with error and may also differ between the parental and the hybrid populations. For example, the protein PRDM9 specifies the locations of recombination hotspots in many mammalian species (reviewed in [39,40]). Traditional methods of intrachromosomal DMI detection are strongly affected by recombination rate variation, whereas in the case of $X(2)$ it solely affects the temporal scale over which DMIs are detected. At the same time, incorporating recombination rate information into DMI detection with $X(2)$ is an exciting direction for future work. That is because if the recombination map was known, our statistics could be incorporated in a simulation-based framework to estimate not only the location but also the strength of selection against the DMI (i.e., the value of the epistasis coefficient $\gamma$).

## Detection time window strongly depends on population demography

The ability to detect a DMI based on negative $X(2)$ and $\Delta D2$ statistics is transient. The time window during which a DMI is detectable with these statistics depends on the demographic history of the population, the strength of the incompatibility, the dominance of the DMI, and the recombination probability between the two loci involved (Figs 2 and S21). Specifically, the time to expose the recombinant imbalance at a DMI increases with decreasing recombination rate between the DMI loci, weaker incompatibility, or weaker dominance of the DMI. Since an

intrachromosomal (or weak interchromosomal) DMI is purged more slowly, it is detectable for a longer time period than an interchromosomal DMI, unless one of the interacting alleles is lost by other forces (such as genetic drift). This interaction between population history and power to detect a DMI must be considered when interpreting $X(2)$ results from real data.

Interestingly, certain migration scenarios improve the power to detect DMIs over long time periods. In isolated hybrid populations, DMIs can only be detected within a limited time window, while the DMI loci are still segregating in the population. With the aid of migration from the minor parental population or balanced migration from both parental populations, DMI loci become monomorphic much later. Specifically, continual immigration from both parental populations can maintain both weak and strong DMIs at a steady state in a hybrid population. Negative $X(2)$ could show these segregating DMIs.

Using the $X(2)$ statistics could be an excellent complementary approach to DMI detection by more established methods. For example, conventional LD-based methods to detect DMIs have moderate power and low false positive rates when admixture did not occur in the last tens of generations and when there is little ongoing gene flow, but they are vulnerable to high false positive rates when there is migration from parental populations [12,13]. Moreover, they are limited to the inference of interchromosomal DMIs in the absence of accurate recombination maps.

In this paper, we have shown that $X(2)$ reliably detects DMIs at the exact position at which the DMI is located. However, we also show that the signal of the DMI loci propagates into their flanking regions. In this vein, the hitchhiking effect is caused by strong DMIs [41]. Even if DMI loci are already monomorphic or at low frequency beyond the detection limit, linked polymorphic loci could still carry the signal of recombinant imbalance. Here we classify this case as a false positive, but in practice considering linked variation could extend the time window for DMI detection. In the future, it would be desirable incorporate linkage information from multiple pairs of negative $X(2)$ between two genomic blocks into our inference method. Since the ancestry tract lengths are negatively correlated with the duration since the onset of hybridization (e.g., [42]), young hybrid populations are then expected to yield strong signals of DMIs across large genomic blocks, whereas older hybrid populations are expected to yield weaker but possibly more finely located DMI signals.

Our study indicates that $X(2)$ can detect DMIs with sample sizes as low as 200 individuals if selection is sufficiently strong and certain demographic conditions are met. In comparison, a previous simulation study, in which the authors compared the performance of various statistical tests to identify DMIs, indicated that DMIs could only be identified with much larger sample sizes (800~1600 individuals suggested, [10]). In the future, phased data or advances in phasing approaches will further improve the power of detecting DMIs with $X(2)$.

## No evidence for enrichment in intrachromosomal DMIs in swordtail population data

Two lines of evidence led us to hypothesize that intrachromosomal DMIs may be overrepresented as compared with interchromosomal DMIs. From first principles, systems and molecular biology have established that genes in the same biological pathway tend to cluster physically in the genome. DMIs may be more likely to arise from disruption or modification of such pathways (reviewed in [14]). This predicts an enrichment in intrachromosomal DMIs under any scenario of speciation. Secondly, evolutionary theory has predicted that tightly linked DMIs are more likely to accumulate and be maintained in the presence of gene flow (e.g., [30]). Thus, both factors should contribute to an enrichment (or reduced purging) of intrachromosomal DMIs. In contrast, in the set of DMI candidates inferred from empirical data in

this paper, interchromosomal DMIs were slightly overrepresented as compared to intrachromosomal DMIs (<4% intrachromosomal DMIs compared with the expectation of 4.2% given an average chromosome length 29.2 ± 4.63 Mb; permutation test, p < 0.02). However, given so many unknowns about the true architecture of selection, we are currently not able to conclude whether this is a biological result or a pattern caused by the limits of our inference method combined with the demographic history of the populations studied here. Although our method is the first that can be applied to detect intrachromosomal DMIs, it is limited to pairs of loci that are at a sufficiently large distance on the chromosome such that recombination occurs with appreciable frequency, thus making the recombinants visible to selection. Therefore, we are likely to miss local interactions, such as those between neighboring genes. Moreover, intrachromosomal DMIs are detected on a different timescale than interchromosomal DMIs, making the two classes not directly comparable. Altogether, our analysis does not allow for a firm statement about the true proportion of intra- to interchromosomal DMIs.

## The challenge of inferring and understanding complex DMIs

We find based on simulations that our statistics are robust to various demographic scenarios such as population expansions and overlapping generations. However, in the modeling and simulation parts of this paper, we mainly study the dynamics of one or two DMI pairs. This is because considering multiple or complex DMIs would generate an intractably large parameter space to explore. Thus, we implicitly assume that DMIs are independent of other incompatible loci in the genome. However, our results indicate that some DMI candidates are involved in many interactions. For example, one locus may interact with multiple loci (e.g., [43,44]). Moreover, DMIs generate fitness costs in hybrids, but these costs can express themselves through different fitness-related phenotypes. This makes it difficult to study DMIs in the same way one would study a quantitative trait. For example, previous studies suggest that *cdkn2a/b* and *Xmrk* interact in some species of swordtail fish (reviewed in [38]). However, *Xmrk* interacts with a previously unidentified locus (*cd97*) to cause melanoma in other swordtail fish species [7]. Our results also point to an interaction between *Xmrk* and a genomic block containing *cdkn2a/b* in swordtail fish, which might cause unfit phenotypes other than melanoma. Gene *rad3d*, at the upstream of *cdkn2a/b*, interacted with *Xmrk* mapped in hybrids between *X. maculatus* and *X. helleri* [45]. Taken together, this suggests the simultaneous presence of multiple hybrid incompatibilities caused by *Xmrk*. This finding is not specific to this study system; recent GWAS and QTL mapping studies in a house mouse hybrid zone also suggested multiple, non-independent genetic incompatibilities [8,46]. In addition to the presence of true hubs of hybrid incompatibilities, other types of selection and interference could contribute to the signal at these loci. For example, there could be interference between different DMIs (analogous to Hill-Roberson interference), or strong negative interactions may cause local stratification of the genome (e.g., [47]).

Moreover, it is important to note that the sensitivity of the $X(2)$ statistic is specific to the recombinant imbalance caused by classical DMIs [48,49]), where only one recombinant haplotype suffers from the incompatibility, whereas the other has similar fitness to the parental genotype. The evolution of this type of hybrid incompatibility is easy to explain without invoking multi-locus interactions [48]. However, hybrid populations may carry hybrid incompatibilities beyond classical DMIs, where both recombinant haplotypes suffer from reduced fitness (e.g., [50]). These tend to be less easily purged by gene flow [30,51,52], and they may cause distinct patterns in the genome such as inflated single-locus heterozygosity. To date, we are far from understanding the relative prevalence of hybrid incompatibilities of different types and their overall role during hybridization and speciation.

According to our DMI inference from hybrid data in swordtail fish, most genomic regions we analyzed are predicted to be involved in DMIs. Although possible, this result is likely an overestimate due to linkage effects and analysis choices made for these data. Specifically, based on our simulations, we observe four phenomena that generate negative $X(2)$: true DMI interactions, interaction signals from a flanking region with a DMI locus, interaction signals from two flanking regions, and interference between DMIs. However, we suspect that we can be confident in loci which we inferred to be involved in DMIs in multiple populations or interactions. Based on this, we speculate that we may have detected 50–70 loci involved in DMIs per swordtail fish hybrid population. In other species, at least 100 loci were experimentally identified among closely related species (reviewed in [53–56]). Previous genetic mapping studies and our study of natural hybrid populations have inferred DMIs at the resolution of large genomic blocks. Because speciation genes and genes involved in the same pathway are expected to cluster, the actual number of incompatible genes might be larger than these current estimates.

## Concluding remarks

In this paper, we take a step towards genome-wide identification of inter- and intrachromosomal DMIs by providing a powerful new tool for researchers in speciation genetics. However, the complexity of potential interactions in the genotype-fitness map of an organism is vast [57], and DMIs are only one specific kind of pairwise epistatic interaction that is characterized by the fitness deficit of one recombinant genotype. Even if this type of (negative) epistasis between two loci is inferred, drawing conclusions about the functional underpinnings of this epistasis is difficult. For example, theoretical work recently demonstrated how epistasis can propagate across modules of a metabolic network, with negative epistasis at lower levels possibly resulting in amplified negative epistasis at a higher level, and positive epistasis at a lower level possibly turning into negative epistasis at a higher level [58]. These findings reinforce the expectation of widespread negative epistasis (including DMIs) for fitness. Mapping and functionally understanding complex interactions and their consequences for genome evolution will require new interdisciplinary approaches that combine the tools and expertise from genomics, systems and molecular biology and population genetics in the future.

## Materials and methods

### A toy model illustrates DMI dynamics

We first consider a toy model of a single DMI pair, similar to [30], that is purged from an isolated hybrid population. Consider two divergent loci resulting in parental haplotypes $aB$ and $Ab$ (S1 Fig). Hybridization and recombination generate two recombinant haplotypes, $ab$ and $AB$. We assume a DMI between $A$ and $B$, and that direct selection acts on the incompatible alleles $A$ and $B$. We denote the fitness of the haplotypes as $w_{ab} = 1$, $w_{Ab} = 1+\alpha$, $w_{aB} = 1+\beta$, $w_{AB} = (1+\alpha)(1+\beta)(1+\gamma)$. Throughout the manuscript, we usually set these coefficients to $\alpha = 0.001$, $\beta = 0.002$, $\gamma = -0.5$, representing weak direct selection and a strong DMI. To gain an intuition of the purging process and the statistics, we first compare the frequency dynamics with and without DMIs by assuming haploid hybrid populations with recombination in a deterministic model.

### Dynamics of the variance in heterozygosity for a two-locus DMI

In the model described above, we denote $g_{ik}$ as the frequency of haplotype $ik$, where $i\in\{A, a\}$ and $k\in\{B, b\}$. The allele frequency of allele $i$ at locus **A** is $p_i$, and the allele frequency of allele $k$

at locus **B** is $p_k$. The heterozygosity $h$ at a locus is $h_A = 1 - p_a^2 - p_A^2$, $h_B = 1 - p_b^2 - p_B^2$. The heterozygosity $h$ is the expectation of the number of heterozygous loci per individual across two loci, $K$, where $K = 0,1,2$. This can be written as the sum of the heterozygosity at **A** and **B** [31],

$$h = h_A + h_B.$$

We define the variance of $K$ as in [26], summing over the contributions of numbers of heterozygous loci (see S10 Table), as

$$\begin{aligned}
\sigma_{het} &= (0-h)^2(g_{Ab}^2 + g_{aB}^2 + g_{ab}^2 + g_{AB}^2) + (1-h)^2(2g_{ab}g_{Ab} + 2g_{ab}g_{aB} + 2g_{AB}g_{Ab} + 2g_{AB}g_{aB}) \\
&\quad + (2-h)^2(2g_{aB}g_{Ab} + 2g_{ab}g_{AB}) \\
&= h_A + h_B - h_A^2 - h_B^2 \\
&\quad + 2\sum_i\sum_k(g_{ik}^2 - p_i^2 p_k^2) \\
&= h_A + h_B - h_A^2 - h_B^2 + 4\sum_i\sum_k p_i p_k D_{ik} + 2\sum_i\sum_k D_{ik}^2,
\end{aligned}$$

where $D_{ik} = g_{ik} - p_i p_k$ is the LD for each haplotype.

If there is no LD, the alleles that segregate at different loci in an individual are independent, so the expected value of the variance of $K$ is $\sigma_2^{exp} = h_A + h_B - h_A^2 - h_B^2$.

The deviation of the variance from the independent expectation is therefore

$$\Delta_2 = 4\sum_i\sum_k p_i p_k D_{ik} + 2\sum_i\sum_k D_{ik}^2.$$

Its relative deviation from the expectation is

$$X(2) = \frac{\Delta_2}{\sigma_2^{exp}},$$

which is the focal statistic of this paper. Similar to LD, $X(2)$ indicates the association between alleles at two loci. The association is created by recombination, genetic drift, and negative epistasis between alleles.

The LD of recombinants is $D = g_{ab} \cdot g_{AB} - g_{aB} \cdot g_{Ab}$.

Normalizing $D$ to $D'$ we obtain

$$D' = \frac{D}{|D_{max}|},$$

where

$D_{max} = max(-p_A \cdot p_B, -p_a \cdot p_b)$ when $D < 0$;
$D_{max} = min(p_A \cdot p_b, p_a \cdot p_B)$ when $D > 0$.
We use D' as a classical measure of LD throughout the manuscript.

## $\Delta D2$ quantifies imbalanced segregation of recombinant haplotypes

We can rewrite the deviation of the variance of $K$ from its expectation in the form of haplotype frequencies ($g_{ik}$) and allele frequencies ($p_i$, $p_k$) as

$$\Delta_2 = 2\sum_i\sum_k(g_{ik}^2 - p_i^2 p_k^2)$$

We propose $D2_{ik} := g_{ik}^2 - p_i^2 p_k^2$ as a measure of the imbalance of haplotype frequencies, which is a key indicator of a DMI.

In a deterministic model without selection, we expect the recombinant haplotypes to be generated at the same frequency after hybridization. It follows that $D2_{ab}$ and $D2_{AB}$ of

recombinant haplotypes are expected to be equal initially, independent of the initial admixture proportions. When there is a DMI, the recombinant haplotype *AB* is depleted, and this expectation does not hold. We use the *D2* difference between the two recombinants to refer to DMI-indicating recombinant imbalance and from now on refer to this as $\Delta D2 := D2_{ab} - D2_{AB}$ (S4A–S4D Fig).

$D2_{ik}$ can also be written as linkage disequilibrium $D$ times a factor, $D2_{ik} = (g_{ik}-p_ip_k)(g_{ik}+p_ip_k) = D \cdot (g_{ik}+p_ip_k)$. Therefore, we expect $D2_{ik}$ to be 0 when the population approaches linkage equilibrium ($D$ approaches 0; S4A–S4D Fig). The total deviation of $D2_{ik}$ from 0, $\sum_i\sum_k|D2_{ik}|$, is an indicator of linkage disequilibrium. The proportion of $\Delta D2$ over the sum of all $|D_{ik}|$, $(D2_{ab}-D2_{AB})/\sum_i\sum_k|D2_{ik}|$, can also be written as $-(1-p_A-p_B)$, showing the relationship between the recombinant imbalance and the elimination of the two alleles that interact negatively in the DMI.

## Haploid and diploid populations and the dominance of epistasis in our model

Modeling DMIs in a diploid population is more complicated than suggested by the toy model, which assumes a haploid population with recombination. In a diploid population, the epistatic interaction depends on the dominance of each incompatible allele (S11 Table). To avoid this complexity, we assumed a haploid population with recombination to demonstrate the dynamics of DMI in a deterministic way with a minimal number of parameters. We can understand the haploid model as assuming that selection acts on the gamete pool. Here, heterozygosity is the proportion of heterozygous sites observed when two haplotypes are randomly selected. S21 Fig shows that $X(2)$ trajectories have similar shapes but with different detection time windows ($X(2)<0$) when a DMI is resolved in a recombining haploid population or diploid populations with various dominance scenarios. Because dominant incompatible alleles are exposed to negative epistasis more strongly, higher dominance results in shorter detection windows (see also [24]).

## Simulation of a DMI under Wright-Fisher and Moran models

In our simulations, implemented in SLiM (version 3.5, [34]), we assumed a population size of 5000 diploid individuals, with selection occurring at the juvenile stage, followed by random mating and recombination. For simplicity and computational speed, we ignored mutation (similar to [24]). At the beginning of the simulation, two fully diverged parental populations of adults are combined at admixture proportions $f$ and $1-f$, which mate randomly. When considering immigration, we assume that migrants from one parental population enter the hybrid population at the juvenile stage after selection. Direct fitness is multiplicative and heterozygous-heterozygous individuals do not suffer from the DMI, similar to the "recessive" DMI scenario in [24,59], where the DMI affects homozygous-homozygous genotypes ($w_{AB/AB} = (1+\alpha)^2(1+\beta)^2(1+\gamma)^4$) and homozygous-heterozygous genotypes ($w_{AB/aB} = (1+\alpha)(1+\beta)^2(1+\gamma)^2$ and $w_{AB/Ab} = (1+\alpha)^2(1+\beta)(1+\gamma)^2$). Individuals were excluded from sampling if they were nearly purely parental in their ancestry (i.e., <10% of genomes from either parental population).

To represent the demography of the studied swordtail fish populations that have overlapping generations, we compared the DMI dynamics and the difference in the time period in which DMIs can be detected between a non-overlapping generation model (Wright-Fisher) and an overlapping generation model (Moran), both of which operate in discrete time. In the Moran model, mortality rates for each age determined the age structure [0.2, 0.2, 0.0, 0.0, 0.0, 0.0, 0.25, 0.5, 0.75, 1.0] from age 0 to 9 in every generation. Age 0 is the new generation of juveniles. The probability of death for each individual was determined according to these mortality rates and individuals that experienced death were replaced by juveniles after selection.

## Sensitivity and specificity

Before applying the statistics to real data, we used simulations under a Wright-Fisher model with variable admixture proportions and migration rates to estimate the sensitivity and specificity of our method at detecting DMIs (Fig 2). We designed these simulations to correspond to realistic conditions for its application to swordtail fish data. According to empirical estimates, the migration rate $m$ ranges from $\ll 0.01$ to $\sim 0.05$ in the natural hybrid populations studied here. To roughly explore these dynamics in our simulations, we included no ($m = 0$), intermediate ($m = 0.005$) and strong ($m = 0.01$) migration from minor parental populations, and two admixture proportions $f \in \{0.3, 0.5\}$. We used the proportion of successful DMI detection under our model to estimate sensitivity or the true positive rate. Specificity was estimated by the proportion of pairs that were correctly classified not to be DMIs under a null model of neutral evolution. Since $X(2)$ was always larger than -0.001 in the 600 simulations we performed for neutrally evolving pairs of loci, DMI detection criteria were set to $X(2) < -0.005$ & $D' < 0$. $D' < 0$ ensures that the detected depleted haplotype is a recombinant but not a parental haplotype. The population size was 5000 in each simulation, and the recombination probability was set to 0.5 for each pair of loci. We simulated each scenario 100 times and randomly sampled 300 individuals for further analysis. Sensitivity and specificity were calculated from generation 10 onwards. 95% confidence intervals were estimated using the Wald test [60].

## False and true positive rates with one and two DMI pairs

To generalize the performance analyses, we simulated genomes with one DMI or two DMIs on two or four chromosomes (chromosomes with DMIs are referred to as DMI chromosomes below). In the same genome, two additional neutral chromosomes were simulated to estimate the false positive rate. One hundred markers were simulated on each chromosome, and the population size was set to 5000. The position of the DMI loci was distributed randomly on the DMI chromosomes such that DMIs could be either intra- or interchromosomal. Epistatic coefficients were drawn from a uniform distribution ($\gamma \in [-1, -0.001]$). The direct selection coefficients $\alpha$ and $\beta$ were drawn from an exponential distribution with mean 0.001. Here, the fitness was computed as the product of the respective finesses of the two DMIs. Due to the large parameter space, we only considered three demographic scenarios, equal admixture proportions without migration ($f = 0.5$, $m1$, $m2 = 0$), equal admixture proportion with migration from both parental population ($f = 0.5$, $m1 = m2 = 0.001$) and uneven admixture proportions with high migration rates from the minor parental population ($f = 0.3$, $m = 0.01$). We simulated each scenario 100 times, such that 100 DMIs in the one-DMI scenario and 200 DMIs in the two-DMI scenario could be used to determine the true positive rates. We also studied the distribution of the total number of retained two-locus haplotypes when only homozygous loci were considered for varying sample sizes ($n \in \{100, 200, 300\}$) in the two-DMI model. At least 200 individuals had to be sampled to obtain >50 homozygous-homozygous haplotypes (S22 Fig). The mean and 95% confidence interval of the false positive rate was estimated by resampling 2000 interactions from all pairwise statistics of 100 simulations 1000 times.

To represent interaction hubs, we implemented a three-locus two-DMI model that directly extends the two-locus implementation. Here, two loci (**A** and **B**) are interacting with a third locus (**C**). In our implementation, this results in two pairwise DMIs between alleles $A$ and $C$, and $B$ and $C$ (S12 Table). With multiplicative fitness, haplotype $ABC$ then suffers the lowest fitness in the population. Among the three DMI alleles ($A$, $B$ and $C$), allele $C$ on average suffers the lowest marginal fitness because three of the four haplotypes that contain allele $C$ ($AbC$, $aBC$, $ABC$, not $abC$) are under strong negative epistasis. Allele $C$ was thus eliminated very quickly from the population. Here, the true and false positive rates were surveyed under equal

admixture without migration ($f = 0.5$, $m = 0$). We analyzed five chromosomes, including three chromosomes with randomly distributed DMI loci and two neutral chromosomes.

## Fish data availability, quality control and bias control

We analyzed published data from three natural populations (CHAF, TLMC, TOTO). CHAF data was described in [7] and published at https://datadryad.org/stash/dataset/doi:10.5061/dryad.z8w9ghx82; TLMC and TOTO data were described in [18] and published at https://datadryad.org/stash/dataset/doi:10.5061%2Fdryad.rd28k4r (S4 Table). Previous work has described our approach for local ancestry inference in *X. birchmanni* x *X. malinche* hybrids in detail [7,61]. Ancestry informative sites were defined based on a combination of low and high coverage sequenced individuals from two allopatric *X. birchmanni* (N = 150) and two allopatric *X. malinche* populations (N = 33). The difference in sequencing effort reflects a large difference in genetic diversity and the expected number of segregating sites between the two species (*X. birchmanni* π per basepair is 0.1%, *X. malinche* is 0.03%). Based on observed allele frequencies at these sites, we excluded any ancestry informative site that had less than a 98% frequency difference between the two samples. We used observed frequencies at these ancestry informative sites, in combination with counts for each allele from low-coverage whole genome sequence data to infer local ancestry using a hidden Markov model (HMM) based approach. The expected performance of this approach was evaluated in simulations, on pure parental individuals not used in the definition of ancestry informative sites and on early generation $F_1$ and $F_2$ hybrids where small switches in ancestry can confidently be inferred to be errors. A subset of simulations also included genetic drift between the reference panels and parental populations that form the hybrid populations [61]. Based on these analyses we estimated error rates to be ≤0.2% errors per ancestry informative site in all tested scenarios [7,61]. Thus, given the high expected accuracy of ancestry inference in *X. birchmanni* x *X. malinche* hybrids, we do not expect that errors will cause the patterns observed in the empirical data from natural populations.

For convenience in downstream analysis, posterior probabilities from the HMM for ancestry state were converted to hard calls. Sites with a posterior probability greater than >0.95 for a particular ancestry state (homozygous *X. malinche*, heterozygous, homozygous *X. birchmanni*) were converted to a hard call for that ancestry state. This resulted in 629112, 629582, and 629101 calls at ancestry informative sites for analysis across populations. Individuals were excluded from this dataset if they were nearly pure parental in their ancestry (i.e., <10% of genomes from either parental population).

To avoid biases, we filtered ancestry informative sites (markers) in the three populations independently using the same criteria. Here, we defined as homozygous genotype when the same ancestry state is at one ancestry informative site in one individual; as heterozygous genotype when the different ancestry state is at one ancestry informative site in one individual. In each population, we retained a locus for analysis if its frequency of each homozygous genotype was >0.05, and if the locus was missing (or heterozygous) in fewer than 60% of individuals. Ancestry informative sites were thinned by LD ($r^2$>0.9) within a 10kb window, with 1-marker step size by PLINK (v1.9, [62]). The thinning resulted in 18,188, 7,244, and 7,415 markers for CHAF, TLMC, TOTO respectively.

We then sampled 700 markers and calculated the statistics (including $X(2)$, $D'$, $\Delta D2$ and haplotype frequencies) pairwise in this data set. If $X(2)$<−0.005 and $D'$<0 (S1–S3 Tables), we considered a pair of markers as a candidate DMI pair. To compare the DMI locations among bootstrap replicates and between hybrid populations, we next split the genome into 1Mb bins (based on the reference genome of *X. maculatus*; [63]). This window size was inspired by our

simulations and two examples in three hybrid populations, which showed that the signal of selection at DMI loci could extend to at least 1-2Mb (Figs 3, S6, S18 and S19). Any two bins that contained pairwise interaction candidates were finally considered a putative DMI. We repeated this procedure 20 times.

The proportion of putative DMIs in all pairwise comparisons was very similar to the expected false positive rate obtained from our simulations, and the proportion of putative DMIs was elevated when markers were assigned to 1Mb bins (S3 and S5 Tables). To be conservative, given this result, we used several approaches to gain confidence in our DMI candidates. We first bootstrapped 20 times for each thinned dataset. Each bootstrap sample contained 700 markers. We used the median proportion $p_{bin}$ of candidate bins with negative $X(2)$ over the 20 bootstrap samples as a baseline false positive rate. We then computed the probability of sampling the same candidate interaction between bins $n$ times (based on choosing different markers in each round); this number should be binomially distributed with parameter $p_{bin}$. Using a binomial test, we determined $k$ as the least number of times that an interaction candidate should be observed to be retained as a putative DMI. For this, we used a p-value threshold corrected for an arbitrarily determined false discovery rate of 0.001 (following the Benjamini-Hochberg procedure, S5 Table), such that the p-value is smaller than (# of putative DMI pairs at the p-value threshold)/(total number of significant pairs in 20 bootstrap samples)·0.001. We obtained values of $k$ between 6 and 10 for the three hybrid populations (see S5 Table) when $X(2) < -0.005$. As an additional approach to reduce false positives, we also limited our focus to interactions that were observed in multiple populations. Independent observation of interactions in two or more populations provides strong evidence that these interactions are not expected by chance.

## Supporting information

**S1 Fig. Illustration of the genomic composition and the fitnesses in the haploid model.** A. In our toy model of hybridization with a single DMI pair, two recombinant haplotypes (*ab* and *AB*) are generated from the parental haplotypes (*Ab* and *aB*) from the F2 onwards. In the following generations, all four haplotypes are segregating. In the Dobzhansky-Muller incompatibility (DMI) model, the recombinant haplotype *AB* is strongly selected against. Throughout the paper we allow for weak direct selection for the incompatible alleles (here, $\alpha = 0.001$ for *A* and $\beta = 0.002$ for *B*) and an intermediate to strong fitness interaction between *A* and *B* (here, $\gamma = -0.5$). B. Recombinant imbalance caused by a DMI as compared with Hill-Robertson interference (HRI). Alleles *A* and *B* under strong selection in the same direction can lead to recombinant imbalance between *ab* and *AB*. The arrows show the intensity of the absolute reduction of each haplotype frequency in one generation.
(TIFF)

**S2 Fig. Deterministic frequency dynamics of the four haplotypes in a new hybrid population in the deterministic haploid model.** Alleles *A* and *B* are under weak direct selection ($\alpha = 0.001$ for *A* and $\beta = 0.002$ for *B*). The epistatic interaction between alleles *A* and *B* is $\gamma$. The recombination probability $c$ is 0.1. The admixture proportion is 0.5. A. The haplotype frequencies approach linkage equilibrium soon after hybridization without a DMI ($\gamma = 0$). B. With a strong DMI ($\gamma = -0.5$) the emerging recombinant haplotype *AB* is eliminated quickly, whereas the *ab* haplotype has a marginal advantage that drives it to high frequencies.
(TIFF)

**S3 Fig. Haplotype frequencies that create negative $X(2)$.** The three panels represent three different admixture proportions for *Ab* and *aB*. The frequencies of four haplotypes ($f_{aB}, f_{Ab}, f_{ab}$,

$f_{AB}$) were plotted. Haplotype frequency of $ab$ ($f_{ab}$) is shown on the x axis and the frequencies of the other three haplotypes are shown on the y axis. The ratio of $f_{aB}$ and $f_{Ab}$ stays the same as their initial ratio for every combination of haplotype frequencies ($f_{aB}, f_{Ab}, f_{ab}, f_{AB}$). The slope and intercept of the grey diagonal line are 1 and (0,0). As the combination of four haplotype frequencies always lie under the grey line, the haplotype $ab$ has always the largest frequency. The haplotype $AB$ has always the smallest frequency. In this figure, $f_{ab}$ could be close to but smaller than 1.
(TIFF)

**S4 Fig. Dynamics of $D2_{ik}$ and $X(2)$.** A-D. $D2_{ik}$ in four scenarios combining equal/unequal admixture proportions and with/without DMIs, obtained from the deterministic haploid model. E-F. $X(2)$ trajectories with (blue dashed line) and without (red line) DMIs. The direct selection coefficients are $\alpha$ = 0.001, $\beta$ = 0.002. The strength of epistasis is $\gamma$ = -0.5 (strong DMI) and $\gamma$ = 0 (no DMI). The recombination probability is 0.1.
(TIFF)

**S5 Fig. Comparison of the dynamics of $X(2)$ and $\Delta D2 = D2_{ab} - D2_{AB}$ for a DMI (A&B) compared with strong direct selection (Hill-Roberson interference, HRI, C&D) in the deterministic haploid model.** The direct selection coefficients acting on $A$ and $B$ are $\alpha, \beta$. The admixture proportion is $f$ = 0.5. Black dots highlight the generation with the largest $-X(2)$ or $-\Delta D2$ for each trajectory. The values of $X(2)$ and $\Delta D2$ are represented by blue color intensity. Blank areas represent either $X(2) > 0$ (to the left of the blue area) or at least one locus close to fixation (minor allele frequency < 0.005, on the right of the blue area). The recombination probability between the two DMI loci is 0.1. A&B. DMIs ($\alpha$ = 0.001, $\beta$ = 0.002, $\gamma < 0$ varies on the y axis) exist for a longer longer time in the population than HRI (C&D). C&D. HRI ($\gamma$ = 0, $\alpha$&$\beta < 0$, y axis) can briefly induce weak negative $X(2)$ and $\Delta D2_{ik}$ earlier than a DMI, but the two-locus polymorphism is quickly purged from the population.
(TIFF)

**S6 Fig. Negative $X(2)$ pinpoints an intrachromosomal DMI in simulated data.** The distribution of $X(2)$, $D'$, and $\Delta D2$ of a chromosome with a DMI pair at generation 50 under a Wright-Fisher model for one simulation run using SLiM. The direct selection coefficients are $\alpha$ = 0.001, $\beta$ = 0.002. The strength of epistasis is $\gamma$ = −0.5. The interacting loci generating the DMI were located at chromosome 1 (chr1):15 cM and chr2: 15 cM, such that the recombination probability was $c$ = 0.5 between the two loci. A-C. Heatmaps of $D'$, $X(2)$ and the $\Delta D2$ within the simulated chromosome at generation 20 (G20). D-F. Heatmaps of $D'$, $X(2)$ and the $\Delta D2$ within the simulated chromosome at generation 50 (G50). Only the DMI loci and their 10 cM flanking regions were shown in D-F. Black arrows highlight the location of the DMI.
(TIFF)

**S7 Fig. Negative X(2) dynamics for a DMI persist for long times if there is ongoing immigration from both parental populations.** Each line is one simulation run using SLiM, assuming a strong DMI ($\alpha$ = 0.001, $\beta$ = 0.002, $\gamma$ = −0.5) under a Wright-Fisher model. The recombination probability between the interacting loci was 0.1. 300 individuals were sampled in each simulation. Trajectories were plotted from generation 2 onwards. DMIs can be maintained by symmetric migration from two parental populations ($m1$, $m2$ = 0.01) for a long time, and are not affected by the initial admixture proportions (Admix. Pro.).
(TIFF)

**S8 Fig. Proportion of hybrid individuals under five demographic scenarios.** Here, hybrids are defined as individuals that derived at least 10% of their genome from each parental

population. Each line is one simulation run using SLiM, assuming equal admixture proportions of the parental populations and a strong DMI ($\alpha = 0.001$, $\beta = 0.002$, $\gamma = -0.5$) under a Wright-Fisher model. The recombination probability between the interacting loci was 0.1. Three hundred individuals were sampled in each simulation.
(TIFF)

**S9 Fig. Dynamics of haplotype frequencies and *X*(2) for a DMI under diploid Wright-Fisher (WF; dash-dotted line) and Moran (solid line) models for one simulation run using SLiM, assuming a population size of 5000.** The model parameters were $\alpha = 0.001$, $\beta = 0.002$, $\gamma = -0.5$, with recombination probability $c = 0.1$ between the DMI loci. Three hundred individuals were sampled in each simulation. A. Negative *X*(2) appears ~50 generations earlier under a Wright-Fisher model than under a Moran model. B. Haplotype frequencies change more quickly in the Wright-Fisher model than that in the Moran model. Lines indicate the haplotype frequencies of *AB* (red), *ab* (black), *Ab* (blue) and *aB* (green).
(TIFF)

**S10 Fig. Sensitivity and specificity of negative *X*(2) for detecting an intragenic DMI with *c* = 0.1, computed from 100 simulations of each parameter combination using SLiM, assuming a Wright-Fisher population of 5000 individuals.** We considered two admixture proportions, $f = 0.5$ and $f = 0.3$, and three migration rates: 0, 0.01, and 0.05. A strong DMI ($\alpha = 0.001$, $\beta = 0.002$, $\gamma = -0.5$) and no DMI ($\alpha = 0.001$, $\beta = 0.002$, $\gamma = 0$) were simulated under a Wright-Fisher model. The population size was 5000. Here, any interaction with $X(2) < -0.005$ & $D' < 0$ was taken as a putative DMI. Trajectories ended (indicated by large plot markers) when one of the interacting loci became monomorphic (A&B) or there were <50 homozygous genotypes for statistics (C&D). 300 individuals were sampled in each simulation. Vertical lines indicate the 95% confidence interval (Wald method). A&B Sensitivity and specificity using phased data. C & D. Sensitivity and specificity when only homozygous genotypes were used at each locus. A&C. Sensitivity is high unless strong migration leads to swamping of the DMI alleles. B&D. Specificity is high under all tested scenarios. Vertical lines indicate the 95% confidence interval (Wald test).
(TIFF)

**S11 Fig. Sensitivity and specificity of the G test that indicates DMIs (*p*<0.05).** We considered two admixture proportions, $f = 0.5$ and $f = 0.3$, and three migration rates: 0, 0.005, and 0.01. A strong DMI ($\alpha = 0.001$, $\beta = 0.002$, $\gamma = -0.5$) and no DMI ($\alpha = 0.001$, $\beta = 0.002$, $\gamma = 0$) were simulated under a diploid Wright-Fisher model. The recombination probability was 0.5. For each parameter combination, we ran 100 simulations with SLiM. Sensitivity (A) was the proportion of successful DMI detections with the strong DMI. Specificity (B) was the proportion of true negative detections without DMI. Trajectories ended when one of the interacting loci became monomorphic or at generation 150 (indicated by large markers). Vertical lines indicate the 95% confidence interval (Wald test).
(TIFF)

**S12 Fig. The distribution of *X*(2) plotted against p values from the G test.** Data were obtained from 100 simulations of a single DMI under a diploid Wright-Fisher model. Here, we used the same data as in Figs 5 and S11 with equal admixture proportions and no migration within 150 generations, i.e., the data points represent the *X*(2) and G test statistics at the DMI loci, pooled across all sampling points. A strong DMI ($\alpha = 0.001$, $\beta = 0.002$, $\gamma = -0.5$) was simulated under a diploid Wright-Fisher model. The recombination probability was 0.5. The distribution was zoomed into two areas by two independent criteria for detecting DMI: (A) small p values by G test ($p < 0.05$); (B) negative *X*(2) values. Red dots indicate DMIs that are observed

before Generation 5 with a significant p value by the G test in panel A. Black dots indicate DMIs after Generation 5 in panel A and B. Only 9.9% of the simulations showed a significant p value by the G test ($p<0.05$) in panel B.
(TIFF)

**S13 Fig.** Sensitivity and specificity of negative $X(2)$ (A&B) and G test (C&D) indicating DMIs when only homozygous genotypes were used at each locus. We considered two admixture proportions, $f = 0.5$ and $f = 0.3$ and three migration rates ($m$): 0, 0.005 and 0.01. A strong DMI ($\alpha = 0.001$, $\beta = 0.002$, $\gamma = -0.5$) for sensitivity and no DMI ($\alpha = 0.001$, $\beta = 0.002$, $\gamma = 0$) for specificity were simulated under a diploid Wright-Fisher model. The recombination probability was 0.5. For each parameter combination, we ran 100 simulations using SLiM. In panel A&B, any interaction with $X(2)<-0.005$ & $D'<0$ was taken as a putative DMI. In panel C&D, any interaction with $p<0.05$ was taken as a putative DMI. Sensitivity (A&C) was the proportion of successful DMI detections with the strong DMI. Specificity (B&D) was the proportion of true negative detections without DMI. Trajectories end when there were <50 homozygous genotypes for statistics or at generation 150 (indicated by large markers). Vertical lines indicate the 95% confidence interval (Wald test).
(TIFF)

**S14 Fig. Distributions of negative $X(2)$ when $D'<0$ for DMI detection under four demographic scenarios.** This figure represents two starting frequencies of the minor parental population, $f = 0.5$ and $f = 0.3$, and different immigration rates from the two parental populations ($m1$ from minor parental population, $m2$ from major parental population). Light green boxes show the $X(2)$ distribution for the three-locus two-DMI model (3-locus) and the other boxes for the four-locus two-DMI model. DMIs were randomly distributed on four chromosomes. Direct selection coefficients on two incompatible alleles were drawn from an exponential distribution with mean = 0.001. Epistatic coefficients were drawn from a uniform distribution [-1, -0.001]. We calculated $X(2)$ statistics using homozygous genotypes from a sample of two hundred individuals for each simulation. We simulated each scenario for 100 times with SLiM. $X(2)$ values were summarized for three types of genomic regions, neutral chromosomes (A), chromosomes with DMIs (B), and individual DMI loci (C). Boxplots represent the interquartile range; whiskers extend to minimal and maximal values. The dash line is $X(2) = -0.005$.
(TIFF)

**S15 Fig. Propagation of the DMI signal into flanking regions in simulated data (sampled at generation 30, significance threshold: $X(2)<-0.005$ & $D'<0$) when only homozygous genotypes were sampled (sampling criteria for panels A&B: homozygous genotype frequency $>0.05$, heterozygosity$<0.6$; panel C shows homozygous data without further filtering).** Data resulted from 100 simulations using SLiM, in which two DMIs (i.e., four DMI loci) were randomly distributed on four chromosomes, thus resulting in inter- and intrachromosomal DMIs. The direct selection coefficients α and β were drawn from exponential distribution with mean value 0.001. The strength of epistasis was drawn from a uniform distribution, γ∈ [-1, -0.001], which is represented by the blue colour intensity in panels B&C. To resemble the demography of the studied hybrid swordtail fish populations, the admixture proportion was set to $f = 0.3$ and the migration rate was $m = 0.01$. 200 individuals were sampled from each simulation to obtain the statistics. A. The number of significant marker pairs decreases rapidly with distance to the nearest true DMI locus. B&C. Proportion of significant marker pairs when statistics were computed for all pairs between two increasingly large regions around the true interacting DMI loci, with (B) and without (C) strict sampling criteria. Red dots indicate the

median obtained from 100 simulations.
(TIFF)

**S16 Fig. The observed occurrence spectra of detected DMI candidates in the hybrid populations is skewed towards low and high frequencies compared to the expected distribution.** The bar plot shows the frequency of occurrence of loci in the putative DMIs for each population. The blue line shows the expected occurrence distribution under a Poisson distribution.
(TIFF)

**S17 Fig. Chromosomal illustration of putative DMIs in three hybrid populations (CHAF, TLMC and TOTO).** Some loci have many putative interacting partners. Each line is colored by the number of putative partners of the two involved candidate loci. In TOTO, loci show a smaller number of interactions than in CHAF and TLMC.
(TIFF)

**S18 Fig. Example of an intrachromosomal DMI candidate in hybrid swordtail fish on chromosome 20.** The values of $X(2)$, linkage disequilibrium ($D$'), and recombinant imbalance ($\Delta D2 = D2_{MB} - D2_{BM}$) are presented by color intensity. As highlighted by an arrow, the candidate DMI was detected in TLMC between locus chr20:9Mb (B alleles) and chr20:10Mb (M alleles). In CHAF, the residual parental linkage is still so strong that $X(2) > 0$, but $\Delta D2$ shows a weak recombinant imbalance. In TOTO, a stronger interaction presumably happened between chr20:8Mb and chr20:11Mb such that we see no significant signal between locus chr20:9Mb and chr20:10Mb.
(PDF)

**S19 Fig. Example of an interchromosomal DMI candidate in hybrid swordtail fish between chromosome 19 and 20.** The values of $X(2)$, linkage disequilibrium ($D$') and recombinant imbalance ($\Delta D2 = D2_{MB} - D2_{BM}$) are presented by color intensity. As highlighted by an arrow, the candidate DMI was inferred in TLMC between locus chr19:8-9Mb (B alleles) and chr20:23Mb (M alleles). We suspect that negative $X(2)$ is due to linkage and that the true DMI locus has been fixed in all three populations. Most of the homozygous M alleles at chr20:22-24Mb have been purged in TOTO but not at chr19:8-9Mb. CHAF shows a larger genomic region involved in DMI than TLMC, which might be due to the recombination map at the first several generations after hybridization.
(TIFF)

**S20 Fig. Scatterplot of $X(2)$ and $D$' for markers occurring at Xmrk and its interacting locus cdkn2a/b.** Markers were extracted from the focal genes and their flanking regions (*cdkn2a/b at* chr5:15.7Mb-16Mb, *Xmrk* at chr21:16.5Mb-17.5Mb). No pairs of markers in TLMC passed the filtering criteria. A large proportion of pairs has $D$'<0 & $X(2)$<0 in both CHAF and TOTO between *Xmrk* and *cdkn2a/b*. Negative $X(2)$ values are larger in TOTO (age, ~35 generations) than in CHAF (age, ~25 generations). In TOTO, $D$' = −1 & $X(2)$<0 indicates the absence of genotypes that are linked to DMIs, possibly due to sampling at low frequencies or completed purging of incompatible alleles. Colouring indicates point density.
(TIFF)

**S21 Fig. Dominance effects on $X(2)$ dynamics of DMIs.** The four DMI scenarios are a DMI in the haploid model, homozygous-homozygous DMI (Homo-homo), homozygous-heterozygous DMI (recessive), and heterozygous-heterozygous DMI (codominant). All trajectories were generated with the same parameters. Incompatible alleles $A$ and $B$ are under weak direct selection ($\alpha = 0.001$ for $A$ and $\beta = 0.002$ for $B$). The strength of the epistatic interaction between alleles $A$ and $B$, $\gamma$, is -0.5. The recombination probability $c$ is 0.1. The admixture

proportion is 0.5.
(TIFF)

**S22 Fig. The number of two-locus homozygous-homozygous haplotypes for different sampling sizes ($n$ = 100, 200, 300), obtained from a set of 100 simulations for each parameter combination using SLiM.** Simulations were performed assuming a diploid Wright-Fisher population with 5000 individuals. In each simulation, two DMIs were randomly distributed on four chromosomes (Chr w DMI). Two additional neutral chromosomes (Chr w/o DMI) were also simulated in the same genome. The number of two-locus homozygous-homozygous haplotypes for each pair loci was counted at generation 30 and 50 (G30 & G50). Neutral chromosomes contain fewer two-locus homozygous-homozygous haplotypes on average than chromosomes with DMIs. Boxes represent the interquartile range, and whiskers extend to the 95% confidence interval.
(TIFF)

**S1 Table. False positive rate under three filtering criteria in simulations of a single DMI.**
(XLSX)

**S2 Table. False positive rate under three filtering criteria in simulations of two DMIs.**
(XLSX)

**S3 Table. False positive rate under three filtering criteria simulations of two DMIs when only homozygous loci are used for analyses.**
(XLSX)

**S4 Table. Characteristics of the three hybrid swordtail fish populations and the corresponding genomic data.**
(XLSX)

**S5 Table. Bootstrapping statistics to define putative DMIs in the three hybrid populations.**
(XLSX)

**S6 Table. 1Mb bins involved in >10 negative interactions.**
(XLSX)

**S7 Table. DMIs supported by >1 hybrid population.**
(XLSX)

**S8 Table. Single interacting bins shared by all three populations.**
(XLSX)

**S9 Table. Summary of $X(2)$ values between *Xmrk* and its interacting genes, *cd97* and *cdkn2a/b*.**
(XLSX)

**S10 Table. The number of heterozygous loci in a pair of haplotypes.**
(XLSX)

**S11 Table. Selection forces (epistasis and direct selection) and dominance at a DMI.** Each field in the table indicates the fitness of the corresponding genotype
(XLSX)

**S12 Table. Parameterization of the two- and three-locus DMI models.**
(XLSX)

## Acknowledgments

We thank Vitor Sousa for extensive feedback and discussions and Alexandre Blanckaert for simulation code and discussions. We are grateful for helpful discussions and comments on the manuscript provided by Adamandia Kapopoulou, the IGC-FCUL-unibe journal club, and Milan Malinsky. We thank Kali Tal for her editorial suggestions. Calculations were performed on UBELIX (http://www.id.unibe.ch/hpc), the HPC cluster at the University of Bern.

## Author Contributions

**Conceptualization:** Juan Li, Molly Schumer, Claudia Bank.

**Funding acquisition:** Claudia Bank.

**Investigation:** Juan Li.

**Methodology:** Juan Li.

**Project administration:** Claudia Bank.

**Resources:** Molly Schumer.

**Supervision:** Claudia Bank.

**Validation:** Juan Li.

**Visualization:** Juan Li, Claudia Bank.

**Writing – original draft:** Juan Li, Claudia Bank.

**Writing – review & editing:** Juan Li, Molly Schumer, Claudia Bank.

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
