## [Decision Letter · Decision Letter 0]

7 Sep 2021

Dear Dr Bank,

Thank you very much for submitting your Research Article entitled 'Inferring inter- and intrachromosomal Dobzhansky-Muller incompatibilities from imbalanced segregation of recombinant haplotypes in hybrid populations' to PLOS Genetics.

The manuscript was fully evaluated at the editorial level and by independent peer reviewers. The reviewers appreciated the attention to an important problem, but raised some substantial concerns about the current manuscript. Based on the reviews, we will not be able to accept this version of the manuscript, but we would be willing to review a much-revised version. We cannot, of course, promise publication at that time.

If you decide to revise the manuscript for further consideration at PLOS Genetics, please aim to resubmit within the next 60 days, unless it will take extra time to address the concerns of the reviewers, in which case we would appreciate an expected resubmission date by email to plosgenetics@plos.org.

[LINK]

We are sorry that we cannot be more positive about your manuscript at this stage. Please do not hesitate to contact us if you have any concerns or questions.

Yours sincerely,

Alex Buerkle

Associate Editor

PLOS Genetics

Kirsten Bomblies

Section Editor: Evolution

PLOS Genetics

This manuscript has been thoroughly reviewed by three referees, each of whom found the manuscript to report interesting results that have the potential to represent a good advance for studying epistasis in hybrids. The reviews identify a few consistent gaps in the presentation. These include requests for more information on the evolutionary histories to which the method is likely applicable and limits to its performance under different histories. Similarly, the reviews include a number of follow-on questions regarding the analyses of the swordtail data. Overall, the reviewers identify many substantial points for additional information about both the statistical methods and the empirical application and offer strong suggestions for how to improve the manuscript.

Reviewer's Responses to Questions

**Comments to the Authors:**

Reviewer #1: Comments on “Inferring inter- and intrachromosomal Dobzhansky-Muller incompatibilities from imbalanced segregation of recombinant haplotypes in hybrid populations” by Li et al., submitted to Plos Genetics

This paper describes, evaluates, and applies two new statistics to detect intra- and interchromosomal DMIs in hybrids. The first statistic, X(2), detects deviation in the variance of two-locus heterozygosity from its expected value: X(2) < 0 indicates the presence of DMIs, especially when LD, measured as D’, is simultaneously negative. The second statistic, D2ik, describes the deviation in frequency of a specific two-locus haplotype relative to its expected frequency based on allele frequencies. It is additionally used to compute DeltaD2ik, the deviation in frequency between two haplotypes, for example between the two possible recombinant haplotypes. The statistics are evaluated deterministically and by stochastic simulations, and then applied to three hybrid populations of swordtail fish, where several DMIs could be detected.

I find the new statistics very interesting and a nice approach to detect epistasis. However, the paper tries to combine two topics: the method development on one side, and the results that come from its application to swordtails on the other side. As both topics are very complex, the results for either remain incomplete. I can thus imagine an improvement of the paper by focusing more on one or the other topic by reducing focus on the other.

The method evaluations, specifically the simulations, were done with settings that explicitly fit the specifics of the swordtail system (e.g., lines 677-678). This is a good idea when the focus of the paper is on swordtails, but less so when the method should be applied to other systems. For example, few other systems will have population sizes of n=5000 or small migration rates <0.01 from the minor parent only. For being an empirical paper, on the other hand, the focus is too much on the methods and the conclusions that are currently drawn from the swordtail results are a bit limited.

Generally, I find it great to have methods like this developed, and it would be nice if other researchers could apply it to their data. For this it would be good though to clearly know about the strengths and limitations of the method under a wider range of realistic scenarios.

Specific comments:

Simulations:

In case that the method should be applied more generally, the performance of the method would need to be assessed under a wider range of conditions.

- Population size: very large population sizes were simulated, which will be rather rare for most real hybrid populations. Is there a minimum population size required? How many samples are recommended to get good estimates for the two statistics?

- Migration: Why only looking at minor-parent migration? What happens, when migration comes from both the minor and major parent: should not both considered together?

- Selection: I was wondering what other forms of selection might result in similar patterns (other than the tested Hill-Robertson interference); for example, stronger selection for either alpha or beta? Currently, alpha and beta were held nearly constant at very similar selection coefficients throughout the simulations (i.e., drawn from exponential distribution with mean 0.001, without separate evaluations).

- Recombination rate: Sensitivity and specificity were only estimated for c=0.5, when I understand this correctly? As the claim is that the method can detect both inter- and intrachromosomal DMIs, it would be good to explore performance for (a few) c<0.5 as well. In simulations (Tables S1-S3), DMI pairs were randomly distributed across chromosomes, but the effect of linkage on performance was not assessed separately (i.e., results were pooled across all possible linkages).

Definitions:

- The paper focuses on DMIs, but as it is acknowledged in the concluding remarks, DMIs are only one kind of epistatic interactions – so the method could in fact detect any other epistatic interaction, including positive epistasis, not only hybrid incompatibilities. Is this correct? If so, it would be good to explicitly mention this at the beginning.

- There is a mismatch in the definition of fitness values in lines 102-103 vs. line 602

- Lines 661-662: “F1 individuals do not suffer from the DMI”: this is an important point that needs to be mentioned much earlier in the manuscript, in my opinion

- I found the usage of the term ‘homozygous’ confusing. Does it mean homozygous within each parental population, but not necessarily in the hybrids? Or homozygosity in the classical sense, i.e., the same two alleles at a single locus in the hybrid population? The data were filtered to only contain homozygous loci for some of the analyses, but then the method should not be able to detect anything.

- The terms ‘migration’ or ‘migration from the minor parent’ seem to be used as synonyms sometimes, please clarify

Application to empirical data:

- It was not fully clear to me whether in the three replicate hybrid populations, migration comes solely from the minor parent, or whether there is additional migration from the major parent.

Other:

- It would be nice to see some example plots for interchromosomal interactions (simulations and empirical data), similar to the ones shown in Figure 3 and Figure S11

- I was wondering whether it makes sense to have D2ik, and/or DeltaD2ik, be normalized to always fall within a specific range of values, independent on allele frequencies

Minor comments:

Lines 33-40: what is this class of methods (classical genetic mapping?), which is contrasted here to the approach in the next paragraph?

Line 71: Typo: double “that”

Lines 94-96: Can you provide a few more details why the statistic can distinguish the two sources of LD?

Line 278: Typo: “X” missing from “(2)”

Lines 319-321: When I read the Figures correctly, sensitivity is reduced for the G test with pseudo-phased data, but increased for the X(2) test (as in Figure S7, not S8, as referred to in the text)

Line 423: sentence ends abruptly

Line 471: So far I can see it, negative X(2) are reported for CHAF and TOTO, but not all three hybrid populations

Figure 6: please indicate confidence intervals or standard deviations or anything like that

Figure S1B: must the selection coefficient for ‘ab’ in the HRI model not be 1?

Figure S8: Legend and caption do not match (black vs. red dots); panels A and B do not seem to show the same data in the range where the two zooms overlap

Figure S11 caption: only recombinant imbalance is shown in the Figure, not parental imbalance

Table S2 caption: I assume these are results for two DMIs, not a single DMI, as currently indicated?

Reviewer #2: review is uploaded as attachment

Reviewer #3: This study proposes two novel statistics for identifying targets of negative epistasis, such as BDMIs, from population genomic data. They propose two statistics, one based on the variance in two-locus heterozygosity and one based on haplotypic imbalance. They use two-locus simulations and forward population genetic simulations to explore the properties and performance of these statistics. Results indicate utility for detecting BDMIs when admixture is recent, and when admixture proportions are fairly even (which may be rare in nature), although the presence of gene flow from the minor population can somewhat ameliorate these limitations. While the temporal horizon of the method tends to be ephemeral, there is a a somewhat longer-lived signal when BDMIs are between loci on the same chromosome. The fairly limited parameter space where the authors' statistics have a signal tempers the impact of the study. But in general, this is high quality work with fairly strong novelty as well, and the paper is well-written. Hence, even if the applicability is somewhat limited, I think the study could merit consideration for PLOS Genetics. However, several key improvements are needed prior to publication, as detailed below.

Major comments:

1. I don't think think the authors have done a good enough job of putting their method in the context of other approaches. Essentially, the introduction now reads like "LD methods are bad, so we'll look for something better". This framing is problematic for a few reasons. * First, the authors overstate the problems with typical LD-based methods.

Line 47: "Though widely used, LD-based methods are known to have an unacceptably high false positive rate and low sensitivity, even when researchers attempt to account for confounders such as hybrid population demographic history (Schumer and Brandvain 2016, Satokangas et al. 2020)."

While advantageous to the present study's narrative, this is a pretty dramatic overstatement and not fully supported by the cited studies. The same goes for similar statements at the beginning of the Discussion. These statements should to be qualfied to say that there can be high false positive rates in certain situations, such as for very recent admixture and ongoing gene flow between source populations (thus generating admixture LD, which has been recognized for some time).

* Second, LD is still an integral aspect of the authors' own study.

* Third and most importantly, there is actually a striking (but presently unacknowledged) complementarity between the authors' approach and typical disequilibrium-based incompatibility scans. The authors' method works well when admixture is recent and/or there is ongoing gene flow. Disequilibrium scans have elevated false positive rates in exactly those scenarios, and instead work best after the initial generations of admixture and without high ongoing levels of gene flow from source populations. The authors' approach is useful for intrachromosomal DMIs, where as other approaches can struggle in this scenario. Framing the fit of the authors' method in this way would seem much more accurate and useful to the field.

2. The authors need to more clearly identify and communicate the demographic parameter space in which their method is applicable. Overall, the authors have done a nice job exploring the impacts of a range of demographic factors on their statistics. However, a concern is that the demonstrated temporal horizon of the statistics is limited. Results from the toy model are mostly shown for just the first 150 generations after admixture, and the SLiM simulations use a nearly-best-case scenario of 50 generations. This all seems well and good if the methods are only intended for swordtail data. But otherwise, it seems like a key question for anyone considering using this approach is whether their population of interest fits into the potentially limited parameter space where the statistics are not merely negative, but produce negative values beyond those expected by chance from the background genomic distribution. My suggestion is that the authors should perform the analyses necessary to more fully express when their methods have power (especially with regard to admixture time), and to communicate the applicability of their statistics more clearly in the Discussion.

3. There is a need for greater statistical rigor in dealing with multiple testing and determining which outliers reject the null hypothesis.

The authors assume only 400 loci per chromosome in simulations. Real data may have dramatically more SNPs than that, even if pruned based on LD, and hence the multiple testing problem will explode. Relatedly:

Line 424: "Although our overall false positive rate is low, given the large number of tests performed in our evaluation of the real data, we can be most confident in putative DMIs that are detected in multiple populations."

The lack of statistical rigor here is concerning. I think it is the responsibility of the authors to use quantitative approaches to determine what is significantly unexpected under the null hypothesis. Deeper consideration of statistical threshold, multiple testing, and significance is needed.

4. Graphical presentation of the empirical analysis is absent from the main paper. I think the authors should find a way to bin their two-dimensional outliers into blocks, and then illustrate each of the resulting pairwise interactions on an image of the genome. This would offer a more intuitive depiction of the results than text and supplemental tables alone, or the limited supplemental figures current present. Such measures would enhance/demonstrate the value of the approaches for other potential users.

Minor comments:

Line 338: "When only homozygous loci (pseudo-phased data) were used for the inference, the true positive rate remained similar, but the false positive rate was elevated to 3.6-6.1%"

Please clarify what's being done here and why the false positive rate goes up.

Line 402: Please clarify if "homozygous markers" are SNPs that appear to be fixed between source species, or otherwise how they were chosen. And please also clarify if you mean that only homozygous hybrid individuals were included for a given locus pair.

Line 510: "Traditional methods of DMI detection are strongly affected by recombination rate variation"

This would be more accurate if "intrachromosomal" was inserted before "DMI".

Line 609: Perhaps just stick with either j or k for the allele at locus B.

Not being especially familiar with this literature, I found that a missing conceptual piece is the authors' specific meaning of two locus heterozygosity.

Line 611: "The expected heterozygosity at two loci is the sum of the heterozygosity at A and B"

What is precisely meant by "heterozygosity at two loci" in this usage? If pA and pB are each 0.5, then hA + hB is 1, even though double-homozygotes for any particular allelic combination would typically exist, and so this does not seem like an intuitive definition of two locus heterozygosity. Further explanation and citation would be helpful, in both the Methods and the Results (readers might go to either section first).

Edits:

Line 34: perhaps "hundreds or thousands" would be more accurate.

Line 45: Corbett-Detig used RIL data from between-population crosses, not population genomic data.

Line 65: don't need "may" twice

Line 415: "numbers of partners" would be more illustrative than "numbers of occurrences"

Line 423: unfinished sentence

**Have all data underlying the figures and results presented in the manuscript been provided?**

Reviewer #1: Yes

Reviewer #2: **No: **Full dataset of computed statistics per bin/marker pair not provided

Reviewer #3: Yes

PLOS authors have the option to publish the peer review history of their article (what does this mean?). If published, this will include your full peer review and any attached files.

Reviewer #1: No

Reviewer #2: **Yes: **Hilde Schneemann

Reviewer #3: No

---

## [Decision Letter · Decision Letter 1]

21 Jan 2022

Dear Dr Bank,

Thank you very much for submitting your Research Article entitled 'Imbalanced segregation of recombinant haplotypes in hybrid populations reveals inter- and intrachromosomal Dobzhansky-Muller incompatibilities' to PLOS Genetics.

The manuscript was fully evaluated at the editorial level and by independent peer reviewers. The reviewers appreciated the attention to an important topic but identified some concerns that we ask you address in a revised manuscript

We therefore ask you to modify the manuscript according to the review recommendations. Your revisions should address the specific points made by each reviewer.

[LINK]

Yours sincerely,

Alex Buerkle

Associate Editor

PLOS Genetics

Kirsten Bomblies

Section Editor: Evolution

PLOS Genetics

This revised manuscript has been evaluated again by the three original reviewers. Each notes and appreciates the improvements to the manuscript. Based on their close reading, the reviewers each make a modest set of suggestions for additional changes to the manuscript. I encourage the authors to consider these in preparing a final version of the manuscript.

Reviewer's Responses to Questions

**Comments to the Authors:**

Reviewer #1: Comments on "Imbalanced segregation of recombinant haplotypes in hybrid populations reveals inter- and intrachromosomal Dobzhansky-Muller incompatibilities" by Li et al., re-submitted to Plos Genetics

This revised version of the manuscript has much improved to the former version and I’m very happy to see that all my previous comments have been thoroughly addressed. In particular, simulations were added to investigate the performance of the method under a wider range of demographic settings, such as small population size and different migration rates from both parental populations, and different recombination rates. In addition, previously unclear formulations have now been adjusted so that they are understandable/unambiguous, and errors have been corrected.

Overall, the manuscript is clear and well-prepared, and presents an important new method to the scientific community. As such I think that it will be a very valuable contribution to Plos Genetics. I have only a few very minor comments left.

Minor comments:

1)

Paragraph "Detection time window strongly depends on population demography" (lines 616-624): I very much like the discussion on the power and limitations of the approach. What I believe is still missing is the effect of dominance (Methods lines 791-793; Figure S21), which does not only affect the detection time window, but also the strength of the signal. I think that just one sentence mentioning this in the Discussion should be sufficient, so that readers will be aware of this limitation.

In Figure S21, it is unclear what h0<

Reviewer #2: I am happy with the changes that the authors made to the manuscript and the additional simulation work they have done. They extended the parameter range of their simulations, which will be helpful for others who wish to apply the method to their own study system. Furthermore, the new results on migration from both parental populations reveals that their method might be able to detect DMIs across longer timeframes than the authors could demonstrate previously. The new simulations with 3-locus DMI pairs give some hints towards how the method behaves for complex DMIs, which seem to be prominent in the swordtail system. The new part of the discussion about the phenomena that generate negative X(2) is also a valuable addition, which helps to interpret the swordtail data. It was nice to see the distribution of putative DMI loci from the swordtail data visualised in Figure S17, and I think perhaps this could be moved to the main text as there is currently no figure for this section.

Regarding my confusion about their choice of a haploid model, I appreciate the clarification they gave about the definition of their statistic in the methods section, and the new supplementary figure and explanation addressing different dominance scenarios.

The new Figure 6 does a good job at visualising the difference in X(2) values for the neutral and DMI(-linked) loci. I did not find the new Figure S3 particularly intuitive to interpret. It might help to explain what the green diagonal line is and to mark out impossible regions of the graph e.g. when frequency of ab is 1, the other frequencies cannot be >0.

Overall, I would recommend the paper for publication in PLOS Genetics.

Minor comments:

L172 – It looks like immigration from the parental population can either widen or narrow the detection window in Fig2B. For example, the box in the top right corner (6,287) to 96,55).

L201 – It would be helpful to add which panels in Fig2 we should look at/compare.

L385 – I think this should be -0.00025. Otherwise it would not make sense that X(2) values for neutral chromosomes are lower than for DMI chromosomes. I think perhaps the confusion came from ln vs log10 in Fig6?

L387 – close to 1 should be in brackets and there is a minus missing.

L409 – It might be good to mention the actual values for the true positive rate as well.

Fig6A – Does it make sense that X(2) is negative and also decreases over time for neutral chromosomes?

L889 – ‘makers’ should be ‘markers’

L1141 – ‘A&C. Sensitivity is high unless strong migration leads to swamping of the DMI alleles.’ In FigS10C the lines without migration seem to show the poorest sensitivity?

Supp Figures - It might be helpful to emphasize the differences between Fig S10, S11 and S13, perhaps using titles on the figures.

Code availability – Perhaps the authors could also provide the link to the original swordtail dataset, which I think is on NCBI?

Reviewer #3: I sympathize that the authors needed to respond to a large number of reviewer requests overall. They have definitely improved their manuscript. However, they are still only part-way there in terms of addressing my major comments:

1. I suggested that there was unacknowledged complementarity between the useful parameter space of the authors' approach and that of typical LD-based approaches. The revised text still does not speak to this complementarity, in that there is no acknowledgement of any parameter space where conventional LD approaches have high power and low false positive rates (i.e. histories where admixture is not very recent, and there is not a high level of ongoing gene flow). I am troubled by the persistent tendency to disparage a set of approaches that may still be our best option for many taxa.

2. I said "The authors need to more clearly identify and communicate the demographic parameter space in which their method is applicable..." The authors have somewhat broadened the set of demographic scenarios in their temporal simulations. Although they don't really address whether a predicted negative value of their metric is strong enough for rigorous BDMI detection, this section is still good enough for me if significance is adequately addressed further down.

A minor note - Figure 2C could be clearer in terms of which population gives/receives migrants.

3. I asked for "greater statistical rigor in dealing with multiple testing and determining which outliers reject the null hypothesis". The authors have made some progress in this area, including by illustrating the distribution of X(2) under neutral and DMI scenarios (Figure 6), and from this analysis they choose a cutoff of X(2) < -0.005 to heuristically define a positive result. However, little has changed in their empirical analysis, and it's still not clear to me whether the "false discovery rate" indicated in the Methods incorporates any consideration of genome-wide multiple testing, in order to yield a meaningful true/false positive probability. If not, then to get around linkage effects, perhaps it would be worth considering how many unique pairs of chromosomes had at least one window pair with X(2) below a cutoff, compared with how many chromosome pairs we'd expect to meet that threshold under the null hypothesis.

Minor notes related to this point:

* In Figure 6, the quantity "-log(-X(2))" is challenging to intuit. I wonder about just plotting X(2) itself on a log-transformed y-axis.

* Further discussion of the number of loci per chromosome would also be helpful. It seems to boil down to a question of the optimal window size in light of the number of generations since admixture.

My original minor comments have been satisfactorily addressed, so it is basically points 1 and 3 above where I still have concerns.

**Have all data underlying the figures and results presented in the manuscript been provided?**

Reviewer #1: **No: **There is no mentioning where data will be available (only code)

Reviewer #2: Yes

Reviewer #3: None

PLOS authors have the option to publish the peer review history of their article (what does this mean?). If published, this will include your full peer review and any attached files.

Reviewer #1: No

Reviewer #2: **Yes: **Hilde Schneemann

Reviewer #3: No

---

## [Editor Report · Decision Letter 2]

25 Feb 2022

Dear Dr Bank,

We are pleased to inform you that your manuscript entitled "Imbalanced segregation of recombinant haplotypes in hybrid populations reveals inter- and intrachromosomal Dobzhansky-Muller incompatibilities" has been editorially accepted for publication in PLOS Genetics. Congratulations!

Yours sincerely,

Alex Buerkle

Associate Editor

PLOS Genetics

Kirsten Bomblies

Section Editor: Evolution

PLOS Genetics

Comments from the reviewers (if applicable):

The authors have done a clear and thorough job of responding to the suggestions from the previous round of review and incorporating changes into the manuscript. I agree that the manuscript has improved as a result and will be pleased to see this work published.

**Data Deposition**

http://datadryad.org/submit?journalID=pgenetics&manu=PGENETICS-D-21-01048R2

**Press Queries**

---

## [Editor Report · Acceptance letter]

21 Mar 2022

PGENETICS-D-21-01048R2 

Imbalanced segregation of recombinant haplotypes in hybrid populations reveals inter- and intrachromosomal Dobzhansky-Muller incompatibilities 

Dear Dr Bank, 

We are pleased to inform you that your manuscript entitled "Imbalanced segregation of recombinant haplotypes in hybrid populations reveals inter- and intrachromosomal Dobzhansky-Muller incompatibilities" has been formally accepted for publication in PLOS Genetics! Your manuscript is now with our production department and you will be notified of the publication date in due course.

With kind regards,

Zsofia Freund

PLOS Genetics

On behalf of:
